# Origin and dispersal history of Hepatitis B virus in Eastern Eurasia

Bing Sun[1,16], Aida Andrades Valtueña [2,16], Arthur Kocher[2,3], Shizhu Gao[4], Chunxiang Li[1], Shuang Fu[1], Fan Zhang [1], Pengcheng Ma[1], Xuan Yang[1], Yulan Qiu[1], Quanchao Zhang[5], Jian Ma [6], Shan Chen[7], Xiaoming Xiao[7], Sodnomjamts Damchaabadgar [8], Fajun Li [9], Alexey Kovalev[10], Chunbai Hu[11], Xianglong Chen[12], Lixin Wang[13], Wenying Li[14], Yawei Zhou[15], Hong Zhu[13], Johannes Krause [2] ✉, Alexander Herbig [2] ✉ & Yinqiu Cui [1] ✉

Hepatitis B virus is a globally distributed pathogen and the history of HBV infection in humans predates 10000 years. However, long-term evolutionary history of HBV in Eastern Eurasia remains elusive. We present 34 ancient HBV genomes dating between approximately 5000 to 400 years ago sourced from 17 sites across Eastern Eurasia. Ten sequences have full coverage, and only two sequences have less than 50% coverage. Our results suggest a potential origin of genotypes B and D in Eastern Asia. We observed a higher level of HBV diversity within Eastern Eurasia compared to Western Eurasia between 5000 and 3000 years ago, characterized by the presence of five different genotypes (A, B, C, D, WENBA), underscoring the significance of human migrations and interactions in the spread of HBV. Our results suggest the possibility of a transition from non-recombinant subgenotypes (B1, B5) to recombinant subgenotypes (B2 - B4). This suggests a shift in epidemiological dynamics within Eastern Eurasia over time. Here, our study elucidates the regional origins of prevalent genotypes and shifts in viral subgenotypes over centuries.

Hepatitis B virus (HBV) belongs to an ancient family of hepatotropic DNA viruses, with origins dating back millions of years[1], and still poses a major health burden to humans nowadays[2,3]. HBV infection can lead to both acute and chronic diseases, elevating the risk of cirrhosis and liver cancer-associated mortality[4–6]. HBV strains have been classified into 10 genotypes (A–J) based on nucleotide differences in their complete genome sequences[7–9]. The distribution of HBV genotypes exhibits similarities among countries within the same geographic region but exhibits marked variations across different parts of the world[10]. While genotypes A and D are globally distributed, genotypes E–J are confined to specific regions and contribute to a smaller proportion of infections worldwide[10–13]. Genotypes B and C are highly

[1]School of Life Sciences, Jilin University, Changchun 130012, China. [2]Department of Archaeogenetics, Max Planck Institute for Evolutionary Anthropology, Leipzig 04103, Germany. [3]Transmission, Infection, Diversification and Evolution Group, Max Planck Institute for the Science of Human History, Jena 07745, Germany. [4]School of Pharmaceutical Sciences, Jilin University, Changchun 130021, China. [5]School of archaeology, Jilin University, Changchun 130021, China. [6]School of Cultural Heritage, Northwest University, Xi'an 710069, China. [7]School of Archaeology and Museology, Liaoning University, Shenyang 110136, China. [8]Institute of Archaeology Mongolian Academy of Sciences, Ulaanbaatar 13330, Mongolia. [9]School of Sociology and Anthropology, Sun Yat-sen University, Guangzhou 510275, China. [10]Department of archaeological heritage preservation, Institute of Archaeology of Russian Academy of Sciences, Moscow 117292, Russia. [11]Institute of Cultural Relics and Archaeology, Inner Mongolia Autonomous Region, Hohhot 010010, China. [12]Institute of Archaeology, Chinese Academy of Social Sciences, Beijing 100101, China. [13]Research Center for Chinese Frontier Archaeology of Jilin University, Jilin University, Changchun 130012, China. [14]Xinjiang Institute of Cultural Relics and Archaeology, Ürümqi 830011, China. [15]School of History, Zhengzhou University, Zhengzhou 450066, China. [16]These authors contributed equally: Bing Sun, Aida Andrades Valtueña. ✉e-mail: krause@eva.mpg.de; alexander_herbig@eva.mpg.de; cuiyq@jlu.edu.cn

prevalent in Asia, accounting for more than 95% of infections. In particular, in China, these genotypes are responsible for 27.9% (genotype B) and 64.4% (genotype C) of HBV infections[9,10,14,15]. Genotype B can be further divided into two groups based on the presence or absence of recombination with genotype C[16]. Genotype F predominates among indigenous populations in South America[17,18], while genotype G infections are primarily reported in the Americas and Europe[19]. This genotype has been shown to descend from the ancient Western Eurasian Neolithic to Bronze Age (WENBA) lineage, and has mostly been identified in patients coinfected with HIV[19]. Genotype I is prevalent in northwestern China, eastern India, Laos, and Vietnam[12,20,21]. Genotype J was initially identified in a Japanese patient with a history of residing in Borneo. It shares the highest sequence similarity with HBV strains infecting gibbons and orangutans in parts of its genome, suggesting a recent HBV transmission event between primates and humans[8].

HBV can be transmitted from mother to child at birth[22] or via infected blood and body fluids, including semen and saliva[23,24]. HBV infects humans and a few other primate species[25]. The major reservoirs of HBV transmissions are individuals with chronic HBV infection[22]. Consequently, the spread of HBV is tightly linked to human migration and, therefore, represents a powerful proxy to study human mobility and interactions[26–28]. Advances in laboratory techniques designed for ancient DNA recovery, coupled with DNA enrichment strategies and next-generation sequencing, have enabled the reconstruction of ancient HBV genomes and the investigation of their evolution through time[28–30]. Ancient DNA sequences offer an invaluable tool in the study of long-term evolution of viruses, providing a genomic snapshot spanning 10000 years[28–30].

The first ancient HBV sequences were published in 2012, demonstrating the feasibility of retrieving HBV DNA from ancient human remains[31]. Two studies published in 2018 identified five sequences that group with non-human primates[29,30]. Kocher et al.[28] reported 78 genomes that group with non-human primates in phylogenetic tree. This now-extinct lineage has been named as the Western Eurasian Neolithic to Bronze Age (WENBA) lineage. This lineage was prevalent in Western Eurasia from approximately 8000 to 3500 years ago before it largely gave way to genotypes A and D. Additionally, it gave rise to a group of rare modern strains classified as genotype G[28]. These ancient HBV genomes, thus, uncovered the previously hidden past diversity of this virus in Western Eurasia[28–33]. Although much progress has been made, with 155 ancient HBV genomes published to date, a substantial majority of these genomes have been retrieved from individuals from Western Eurasia. Only two genomes have been recovered from Eastern Eurasian individuals, 12 from the Americas and one from Africa. This notable bias in sampling constrains our understanding of HBV's dispersal and evolutionary history.

In this study, we address this gap by reconstructing and analyzing 34 complete or partial ancient HBV genomes from present-day China, Mongolia and Russia, dating back between 5000 to 400 years ago. The newly reconstructed ancient HBV genomes suggest Eastern Eurasia as a potential origin for genotypes B and D. The high diversity of HBV in the Xinjiang province underscores the profound impact of human migrations and interactions on the dispersal of HBV. The ancient HBV genomes provide evidence for the dynamic history of HBV in Eastern Eurasia.

## Results

### Screening and genome reconstruction

We screened 869 sequence data sets to detect the presence of HBV DNA, most of which were obtained from teeth. For individuals where teeth were not available, the sequence data were obtained from petrous bones. Our screening revealed reads mapping to HBV in 34 individuals from 17 sites in Eastern Eurasia. None of these human remains exhibited pathological lesions identified through osteological examination (Figs. 1 and 2, Supplementary Fig. S1 and Supplementary

data S1). Among all the positive samples, three (XBQM47, XBQM86, XBQM125) yielded DNA from the petrous bone, while the remaining positive samples originated from teeth (Supplementary data S1). The samples, when aligned using bwa, exhibited varying quantities of reads assigned to HBV, ranging from just one read (MY19) to 7205 reads (XHM18). Combining literature on ancient individuals who carried HBV with radiocarbon dating results from 13 positive individuals, we determined their ages to be approximately 5000 years and 400 years ago, respectively[34–36] (Supplementary Table S1). It is important to note that we cannot assess the ancient damage pattern for the samples with less than 200 reads[37] (see Supplementary Fig. S2). However, reads mapping to the human genome revealed the characteristic pattern of damage expected for ancient DNA (see Supplementary Fig. S2)[30]. To enhance the quality of our dataset, we performed an in-solution capture enrichment for HBV DNA for all the samples with reads assigned to HBV[38,39]. Post-capture, genomic sequences were reconstructed by mapping the reads to an HBV reference sequence (Section 1), resulting in genome coverage ranging from 6.05% to 100%, with an average genomic coverage spanning from 0.08 to 1145-fold. Genome coverage of ten sequences reached 100%, six sequences ranged from 90% to 100%, fourteen sequences ranged from 70% to 90%, and only two remaining sequences resulted in less than 50% coverage. However, for the samples XBQM86 and XHM31, the capture experiment was unsuccessful, leading to a loss of DNA content post-capture compared to its pre-capture state. To ascertain the genotypes, we conducted a competitive mapping using representative genomes for each lineage (Supplementary Section 1) categorizing the 34 ancient HBV genomes into five genotypes (Supplementary data S1). After reconstructing the ancient HBV genomes, previously published methods were employed to evaluate the occurrence of mixed HBV infections in certain individuals. Nine individuals (91KLH18, 98JJLM9, AT19, AT7, FLTM101, FLTM48, MY12, MY17, XN12) were identified as having mixed HBV infections (Supplementary data S2). All samples, except for those subjected to full-UDG treatment or samples with few reads mapping to HBV[40], exhibited clear aDNA damage patterns after capture (Supplementary Fig. S2).

### Phylogenetic analysis

To assess the phylogenetic placement of the new ancient genomes in relation to all currently known HBV diversity, we estimated a maximum likelihood (ML) tree using the newly reconstructed ancient genomes that have over 50% genome coverage and a mean coverage greater than 5x (25 in total). These were combined with published ancient genomes meeting the same coverage standard together with modern human and non-human primate HBV genomes (Supplementary Fig. S3a and Supplementary data S3). As we identified eight individuals with mixed infections, an additional ML tree was constructed for the phylogenetic analysis, excluding these individuals (Supplementary Fig. S3b). The position of the newly reported ancient genomes in the ML tree is consistent with the genotyping results. The genome of XBQM86, recovered from the Quanergou site, represents the second deepest branch in the lineage leading to genotype A. The extremely long branch and relatively basal position of this individual may speak for the presence of unsampled diversity of genotype A in the past. Fifteen of the newly recovered genomes fall within genotype B and are widespread throughout Eastern Asia: 96NVZIM6 (Niuheliang site, northeast China), JHM2098 (Hengshui site, northeast China), SBSM101 (Tiantaijie site, northeast China), TJZM25-2 (Taojiazhai site, northwest China), AT7, AT19, AT24 (Bayanbulag site, south Mongolia), XN12 (Derestuj site, south Russia), XHM12, XHM18 (from Xihe site, northeast China), XBQM47 (Quanergou site, northwest China), FLTM48, FLTM97, FLTM101 (Fuluta site, northeast China), 91KLH18 (Longtoushan site, northeast China). In the sequence identity analysis, all ancient sequences show greater than 97% identity with their best-matched modern B subgenotype sequences. Nevertheless, compared

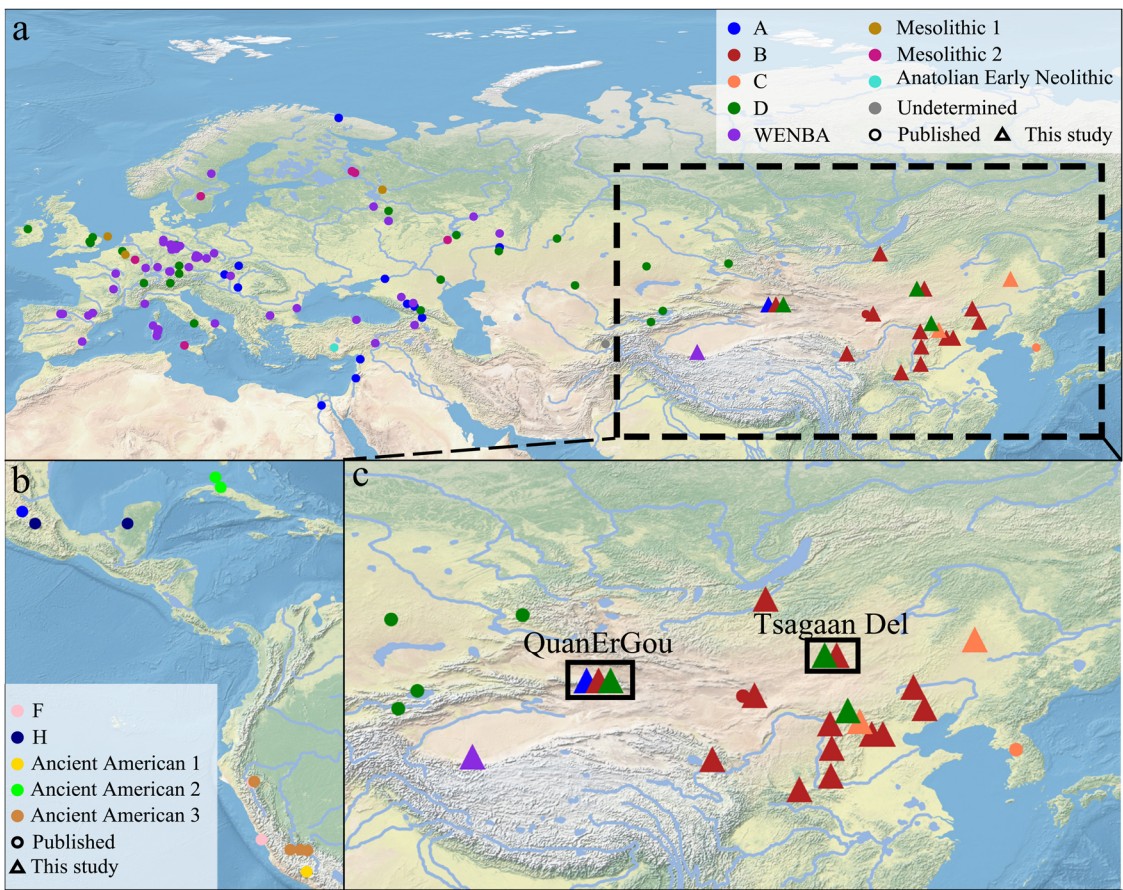

**Fig. 1 | Geographical distribution of ancient individuals with HBV.** Distinct colors represent each genotype, and sites with multiple genotypes present are highlighted with black squares. **a** Geographical distribution and genotype of all the published HBV positive samples and our novel ancient samples. **b** The geographic distribution of various genotypes of published individuals with HBV across the American Continent. **c** An enlarged view of the geographical distribution of ancient individuals with HBV from our study is depicted.

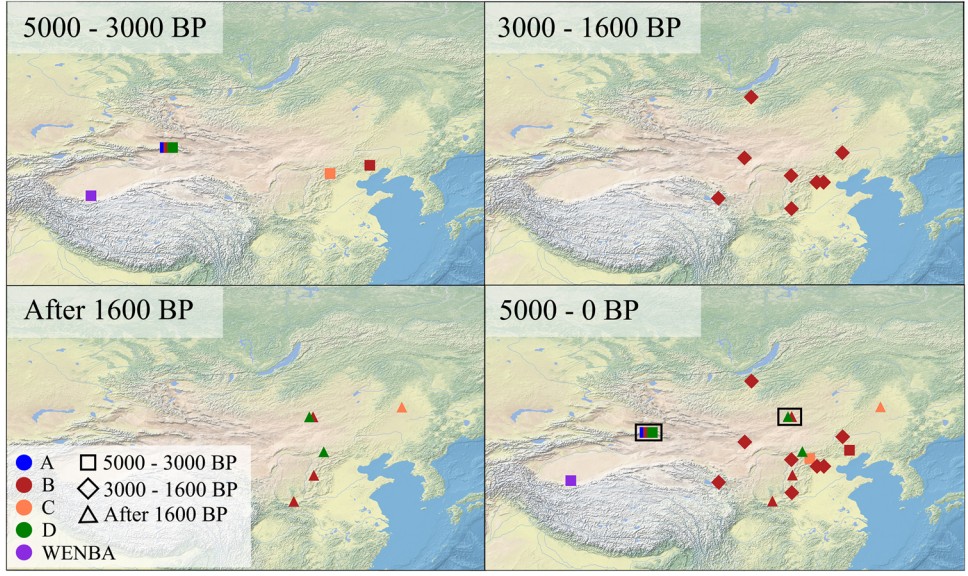

**Fig. 2 | Geographic distribution of ancient HBV genomes within different time-periods.** Black squares highlight sites where multiple genotypes are present. To compare the diversity of ancient HBV across Eastern and Western Eurasia, time intervals were established following the guidelines outlined in Kocher et al.[28].

to modern sequences, these ancient sequences show the highest sequence identity among themselves (Supplementary data S4). Ancient sequences XBQM47, FLTM97, FLTM101, AT7, AT19, AT24, TJZM25_2, XHM12, MY19, XHM23, SBSM101 have the highest sequence identity with modern subgenotype B1 but FLTM101 clusters with subgenotype B5 with a 76% bootstrap value. The ancient sequences 91KLH18, FLTM48, XN12, JHM2098 have the highest sequence identity with modern subgenotype B5 but XN12 and JHM2098 cluster with

subgenotype B1 with 12% and 23% bootstrap value, respectively (Supplementary Fig. S3a). The ancient sequences XHM18 has the same sequence identity with modern subgenotype B1 and B5 (Supplementary Table S2). The 5000-year-old sequence (96NVZIM6) fall basal to all the modern and ancient sequences. Three individuals from a 4130-year-old cemetery in North China are deemed positive for HBV of genotype C. However, only 98JJLM9 (Jiangjialiang site, northeast China) is included in the phylogenetic analysis, which clusters with genotype C. One 400-year-old individual from Honghe site fall basal to all the modern sequences of subgenotype C1. The subgenotype C4, exclusively in indigenous Australians[41], fall basal to all the ancient and modern sequences. 98JJLM9 fall in a lineage placed between subgenotype C4 and other subgenotypes of genotype C. The genomes of MY12, MY17 (Tsagaan Del site, southeast Mongolia), ZQM16 (Qilangshan site, northeast China), XBQM20, XBQM46, and XBQM125 (Quanergou site, northwest China) fall within the diversity of genotype D. Three of them (XBQM20, XBQM46, XBQM125) from the Quanergou site (XBQ site), define a branch that is basal to the entire genotype D lineage. The basal position of XBQ sequences is further confirmed through closer inspection at the nucleotide level, with two unique SNPs shared by these three sequences from the XBQ site. MY17, ZQM16, and BRE008 (published genome recovered from the Hun-Xianbei culture)[28] and DA27 (published genome recovered from the Hun-Sarmatian culture)[30] cluster with modern subgenotype D5. MY12 groups with SHK001, DA222, and MAY017[28–30]. The 11KBM13 (Beifang site, northwest China) genome from the Tarim group[42], clusters with the WENBA lineage, which was widely distributed in Western Eurasia during the Neolithic and Bronze Age periods[28]. This new WENBA genome expands the known geographical spread in which this genotype was present to Eastern Asia.

To infer the time to the most recent common ancestor (tMRCA) of the main HBV lineages, we used the Bayesian framework implemented in BEAST v.2.6.6[43]. To evaluate the presence of a temporal signal in our dataset, we performed a root-to-tip regression test using Tempest with the previously generated ML tree (v.1.5.3)[44]. We observed a good temporal signal in our dataset ($R^2 = 0.7042$) (Supplementary Fig. S4). A dated phylogeny was constructed with BEAST v.2.6.6[43] using two datasets, with or without the mixed infections, identical to those used for the ML tree (Fig. 3 and Supplementary Fig. S5a). In order to choose the most appropriate tree prior and clock model, we performed model selection using path sampling. Both strict and relaxed log-normal molecular clock models were evaluated, incorporating coalescent constant, coalescent exponential, Bayesian skyline and birth death population priors. Model comparisons supported a relaxed log-normal molecular clock model coupled with a coalescent exponential population prior (Supplementary Table S3). The topologies between the ML tree and the Maximum Clade Credibility (MCC) time-tree were mostly consistent, with the exception of different placement within their genotype for RISE387[30], TJZM25-2 (Taojiazhai site), AT7, AT19 (Bayanbulag site), XBQM20, XBQM46, XBQM47, I0216, I0217 (Fig. 3 and Supplementary Fig. S3a). It has been previously reported that recombination with another sequence can affect the topology of the phylogenetic tree[45]. We constructed an unrooted phylogenetic network to provide a clearer visualization of the recombinant nature (Supplementary Fig. S6a, b). We observed low posterior support values for the nodes of the mentioned ancient strains, which could potentially be explained by different phylogenetic placements due to recombination events known to have occurred between all the sequences of modern genotype B and modern and ancient genotype D. The median root age of this resulting tree was inferred to be 13.69 kyr (95% highest posterior density (HPD) interval: 12.104–15.687 kyr) and the median clock rate was $1.375 \times 10^{-5}$ substitutions per site per year (95% HPD interval: $1.249 \times 10^{-5}$–$1.5059 \times 10^{-5}$ substitutions per site per year) (Fig. 3 and Supplementary Fig. S5b), which is in agreement with previous estimates from ancient HBV study[28]. The most recent common

ancestor of genotype A, B, C, D was dated to 6554.8 years old (5857.6–7284.9 y 95% HPD), 5559.8 years old (5114.1–6122.5 y 95% HPD), 5198.4 years old (4647.8–5934.9 y 95% HPD), 4383.9 years old (3806.6–4973.5 y 95% HPD), respectively. The most recent common ancestor (tMRCA) of 11KBM13 (Beifang site) and KAP002 (published genome recovered from a Srubnaya culture)[28] was dated to 4038.3 years ago (3566.0–4598.8 y 95% HPD) (Fig. 3 and Supplementary Fig. S5b).

## Recombination analysis
To investigate recombination events in both ancient and modern HBV, we conducted a recombination analysis with RDP5[46], employing the database used for phylogenetic analysis (Supplementary data S3). Genotype B can be divided into five subgenotypes, of which three are known recombinants (B2–B4)[16]. The ancient genotype B sequences were checked for the presence of recombination with genotype C and no such recombination event was detected (Supplementary Fig. S7a). We determined that subgenotype B2 and B4 are modeled as a recombinant derived from subgenotypes B1 and C2, which served as parental sources and subgenotype B3 was modeled as recombinant derived from subgenotypes B5 and C2 (Supplementary Fig. S7b)[47]. These results are consistent with previous research. Genotype I was modeled as a recombinant derived from subgenotypes A and C (Supplementary data S5). We did not detect recombination events in ancient HBV of genotype B from around 1000 years ago (Supplementary data S6). Due to their lower quality, this does not definitively indicate the absence of recombination. Samples predating 1800 years ago, as well as even older samples, have genome coverages greater than 80%, lending credibility to the authenticity of these results. For samples with low coverage, we performed recombination analysis using SimPlot[48], which also did not detect any recombination events with genotype C, consistent with the results from our RDP5 analysis of all samples (Supplementary Fig. S7a). In our recombination analysis, it was determined that genotype D is modeled as a recombinant derived from genotypes A and WENBA, which served as parental sources (Supplementary Fig. S7b). Additionally, when employing different regions for phylogenetic assessment, the phylogenetic placement of genotype D within the evolutionary tree exhibited shifts (Supplementary data S6).

## Human genomic analysis
In order to understand the difference in the genomic history of the individuals infected with HBV, we performed principal component analysis (PCA) and ADMIXTURE analyses (Fig. 4, Supplementary Fig. S8). In the PCA, principal component one separates East and West Eurasians, and principal component two separates Southern and Northern East Asians. A cline was formed between the Northern Siberian Nganasan population in the top-right of the PCA plot and the indigenous Taiwanese group Ami at the bottom-right (Fig. 4a), with Sino-Tibetan speakers represented by modern Han and Tu, as well as, Tungusic speakers represented by modern Oroqen, Japanese, Korean, and other Eastern Asia populations plotting within this cline. We observed a separation between two groups of individuals infected with genotypes B and D in the PCA plot. Individuals infected with genotype B fall into the cline that includes modern Hezhen, Xibo, Mongolia, Tibetan, Japanese, Korean, and Naxi, with the exception of the individual XBQM47 that represent a nomad-related individual with genotype B (Fig. 4a). The individuals infected with genotype D had a more heterogeneous genetic background, and they were observed in two different clusters (Fig. 4a). In the PCA, one of them was slightly shifted towards Western Eurasians. The position of individuals infected with genotype C shifted slightly towards the direction of Northeast Asians compared to the individuals infected with genotype B (Fig. 4a). These findings are consistent with the ADMIXTURE results. The separation between the individuals infected with genotype B and D was observed

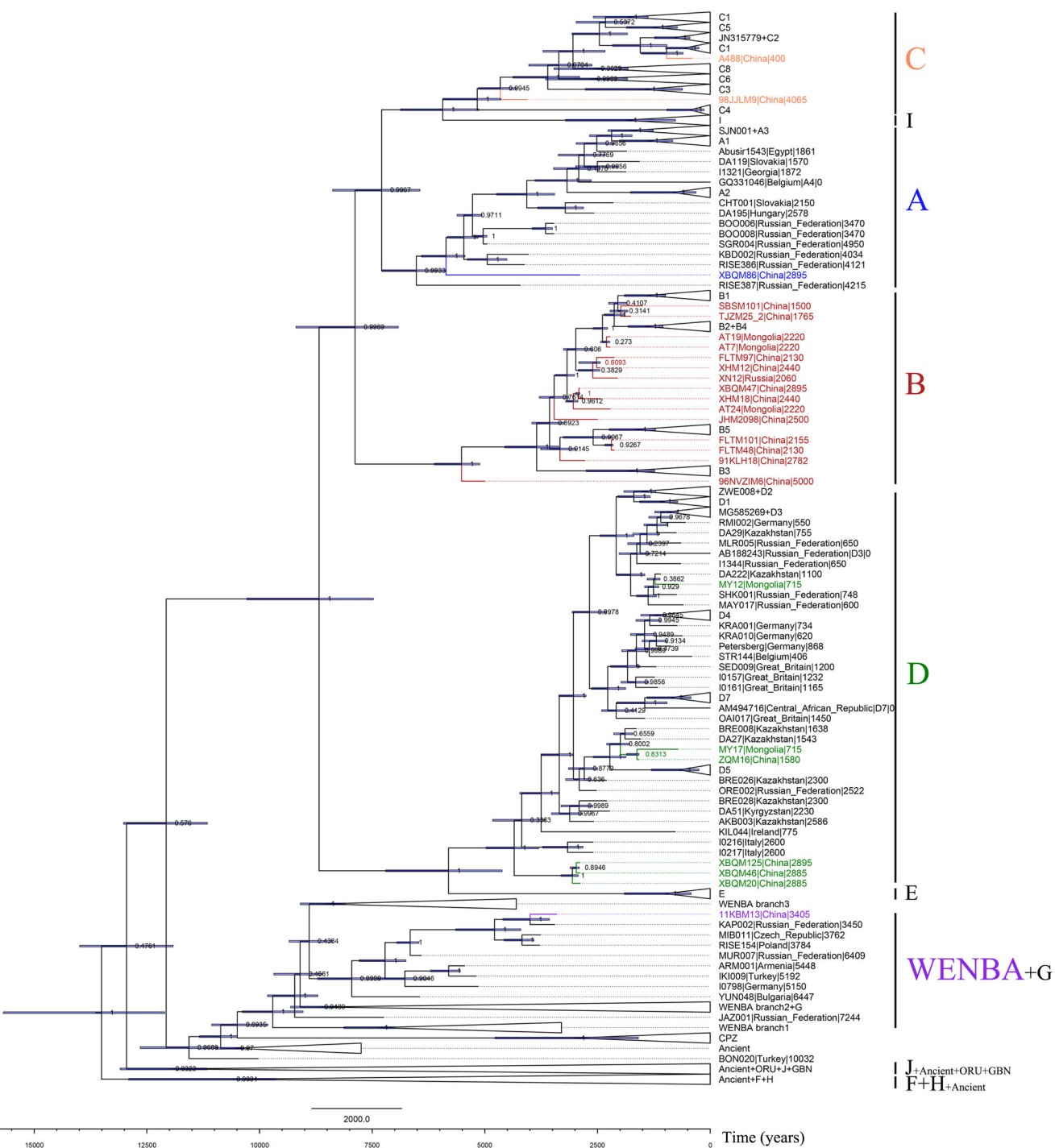

**Fig. 3 | Maximum Clade Credibility time-calibrated phylogenetic tree of modern and ancient HBV.** The dataset used to construct the phylogenetic tree included the mixed HBV infections. The branch lengths represent the sampling time of each genome. The lineages in which our ancient sequences are located are highlighted with different colors. The branches and names of the ancient sequences are also highlighted with the indicated color. The blue horizontal bars represent the 95% confidence interval of tMCRA, and the node labels indicate the posterior values.

in two groups (Fig. 4b). While individuals infected with HBV genotype B shared a similar genetic profile, individuals with genotype D showed different genetic structures (Fig. 4b).

According to the archeological background, DA45 (HBV genome published in 2018)[30,49] and AT19 originate from the same site and these two genomes define a branch with a 100% bootstrap support (Supplementary Fig. S3c). To explore the relationship between DA45 and AT19, we checked the mismatch SNPs of the human DNA of these two individuals and observed that these two samples are from the same

individual (Supplementary Table S2). While the coverage of AT19 was higher (283×) and its library was full-UDG treated, the library of DA45 was No-UDG treated. There were five SNPs that differ between the sequences of DA45 and AT19, with all of them being 'A' in DA45 but 'G' in AT19. Additionally, AT19 displayed 28 SNPs marked as "N" in its sequence due to being mixed. Since the coverage of DA45 (4.3×) is lower, the proportion of mixed sites may differ from AT19 or some mixed sites in DA45 may be undetected. As a result, the data of DA45 and AT19 were not merged, and instead, we substituted the DA45

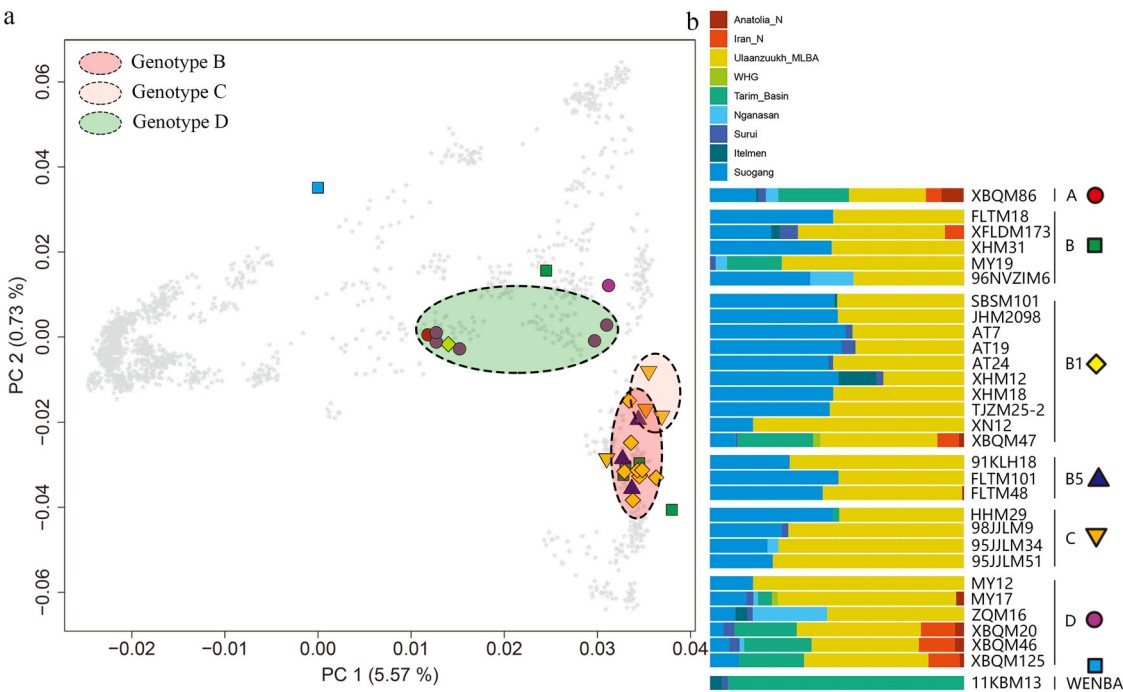

**Fig. 4 | PCA and ADMIXTURE analysis of ancient HBV-positive individuals.**
**a** Principal component analysis of ancient individuals infected with HBV and present-day individuals. The first two principal components (PCs) were constructed from 2077 present-day Eurasians; the ancient individuals are projected onto the first two PCs. Color-filled shapes represent ancient individuals, gray asterisk represents the present-day individuals used for calculating PCs. Colored ovals encompass individuals with the same genotype: green for genotype B, red genotype C and blue genotype D. **b** ADMIXTURE results for the "1240k-Illumina" dataset[104] with K = 9. Based on the genotype and subgenotype of all HBV sequences, we divided them into 7 groups.

sequence with the AT19 sequence in the phylogenetic analysis and recombination analysis.

## Discussion

In this study, 34 ancient HBV genomes were retrieved from human skeletal remains from Eastern Eurasia, providing novel insights into the evolutionary history and geographical origins of HBV genotypes, shedding light on the intricate interplay between disease transmission and human mobility in the past. We found evidence for multiple genotypes present in two of the studied sites: one site located in southeast Mongolia, which was built by Mongol tribes between the 12th–14th century[50], and a second site located in Xinjiang, northwest China (Fig. 2). Our investigation revealed the presence of five distinct genotypes (A, B, C, D and WENBA) within the examined individuals, highlighting the past diversity of HBV in Eastern Eurasia.

Genotype A - D are widely distributed across contemporary Eastern Eurasia and based on our data we demonstrated that they were already present in East Asia as early as 3000 years Before Present (yBP). We also revealed the presence of the WENBA lineage in East Asia, even though genotype G, which descends from WENBA, is presently rare in Asia and remains undetected in China today[10]. This suggests a discrepancy between the distribution of HBV in ancient and modern populations. Compared to Western Eurasia (two genotypes), the HBV diversity at Eastern Eurasia is much higher at this time (five genotypes). All the ancient HBV reconstructed in this study, dating between 3000 - 1600 yBP belong to genotype B, showing the predominant distribution of this genotype in this time period, which is consistent with its high prevalence in modern Eastern Eurasia[10,15]. However, we must acknowledge the potential influence of sampling bias on this pattern. After 1600 yBP, we identified ancient HBV from genotypes B, C and D in this region: three individuals from three different sites carried genotype B, one individual from one site carries genotype C, while three individuals from two sites carry genotype D. Interestingly, we detected B (MY19) and D (MY12 and MY17) from different individuals

of the Tsagaan Del site at the same time, which is attributed to the late Mongol Empire to the Yuan dynasty. Furthermore, our genomic analysis links the detection of genotype D to the ancient Xianbei culture (Qilangshan site, ZQM16)[50–52]. The close relationship observed in the phylogenetic tree between BRE008 (hun-Xianbei)[28], DA27 (hun-sarmatian)[30], SHK001[28], DA222 (karluk)[30], and MAY017 (Golden Horde)[28] with our genotype D individuals is consistent with the cultural interactions of these ancient societies. The reappearance of genotype D may be attributed to the migration of Xianbei populations and Mongols. These snapshots of ancient HBV distribution across various time periods offer valuable insights into the dynamic evolutionary processes that shaped HBV's history.

The observed dynamic distribution of HBV genotypes in ancient Eastern Eurasia raises questions about human population contacts and mobility underlying these patterns. Notably, we found that ancient genomes of genotype B fall into two distinct sublineages and one 5000-year-old sequence fall basal to all the ancient and modern sequences of genotype B. Surprisingly, our human genomic analyses revealed that all individuals carrying genotype B strains shared a remarkably similar genomic profile, indicative of a spread facilitated by population dynamics and migrations. These ancient HBV genomes unveil a rich diversity of genotype B in Eastern Eurasia, dating back 5000 years ago, suggesting a potential origin of genotype B within this region. Compared to the numerous HBV of genotype B we identified, our analysis revealed only one genome of genotype A. Prior studies indicated that oldest ancient sequences of genotype A were recovered from SGR004, RISE386/387 and KBD002. These individuals from western Russia and the northern Caucasus were dated from 5000 to 4000 yBP. In this study, a 2895-year-old sequence from Xinjiang represented the second deepest branch in the lineage leading to genotype A. The presence of several ancient genomes from various locations branching at basal positions within the genotype A lineage challenges our understanding of the geographical origin of this genotype. We've identified 98JJLM9 as the oldest strain of genotype C

recovered so far, showing that the history of genotype C in Eastern Asia dates back more than 4130 years. Furthermore, genotype C is currently the most prevalent genotype in China while its sister clade, genotype I, is currently distributed in China, Laos, and Vietnam[10,15,21]. Collectively, these findings suggest that genotype C has been present in Eastern Eurasia for a long time, and genotype I may have similar ecological adaptability, but the specific reasons require further study. A 3405-year-old individual, from an isolated group in the Tarim Basin, carried the HBV of WENBA. Recent research suggests that the human genetic profile for this isolated group of Tarim formed around 9157 years ago[42]. The tMRCA of the branch formed by 11KBM13 and KAP002[28] was estimated as 4038.3 yBP (3566.0–4598.8 yBP 95% HPD) (Fig. 3 and Supplementary Fig. S5b) and it would speak for a recent introduction with respect to the emergence of this lineage in Europe that has been associated with the early Neolithic 7000–8000 years ago. Certainly, we cannot exclude the possibility that there may exist samples older than 11KBM13 in the region, which could potentially reflect different transmission patterns of WENBA. Interestingly, this individual grouped genetically with the Tarim_EMBA1 in PCA, also supported by the admixture analysis, indicating a lack of admixture with Western Eurasian populations. Nevertheless, Xinjiang shows a rich diversity of economic elements and technologies during that time, like wheat, millet and ephedra twigs, which were originally domesticated in different parts of the world, reflecting the communication of different cultures[36,42,53–56]. All previous WENBA genomes were reconstructed from Western Eurasia. However, given the complex human population history in Xinjiang and the limited number of ancient WENBA sequences from Eastern Eurasia, it is difficult to infer the precise timing and circumstances through which this lineage reached this region.

Moreover, we observed three different genotypes (A, B, and D) present in the Quanergou cemetery. XBQM86 represents the first ancient genome of genotype A recovered from Eastern Eurasia. It forms a phylogenetic branch closely related to Western Eurasian strains, while XBQM47, the westernmost among all ancient genotype B genomes, forms a new branch with XHM18 (Xihe site). The remaining three HBV-positive individuals from this site carried genotype D strains. Xinjiang is located on the Proto-Silk Road, a historic trade route that linked Western and Eastern Eurasia and witnessed the exchanges of people, cultures, agricultural products, and languages[57–61]. Human genomic research on individuals excavated from the Shirenzigou site, located 10 km away from the Quanergou site, suggests that the East-West admixture between Northeast Asian and Yamnaya related populations observed in Xinjiang is more than 2000 years old[62]. Further studies on Bronze and Iron Age populations in Xinjiang reveal a complex demographic history of this region, shaped by the influence of steppe, Central Asian, and East Asian groups over time[63]. The Proto-Silk Road, situated in the heart of Xinjiang, and the resulting high human mobility in this region could potentially have contributed to the spread of HBV, which is further supported by previous research, such as the finding of *Salmonella enterica* in the Quanergou cemetery[64].

Previous analyses suggest that genotype D emerged from recombination between genotype A and WENBA[28](Supplementary Fig. S7a). Our ancient sequences provide the first evidence of geographical overlap of genotypes A, D and WENBA in Xinjiang approximately three thousand years ago. Together with the basal position of these strains in their respective lineages, these findings suggest that genotype D might have originated in this highly interconnected area, potentially facilitating its subsequent spread to other regions. However, we cannot exclude the possibility that this recombination event occurred in another region thousands of years ago, and subsequently spread to Xinjiang.

Recombination is one of the major mechanisms shaping the evolution of viruses, and is known to have played an important role in the evolutionary history of HBV[65,66]. We identified the previously

reported recombinant events involving genotypes B and C and giving rise to subgenotypes B2, B3 and B4. These recombinants lineages originated from two separate recombination events, with their major parent being B1 (B2 and B4) and B5 (B3), respectively (Supplementary Fig. S7a, Supplementary data S5 and data S6). This is also consistent with the patterns observed in the phylogenetic tree. Notably, none of the ancient genotype B samples identified in Eurasia so far exhibit recombination events with genotype C, represented by modern genotypes B1 and B5. Nowadays, non-recombinant B genotypes (B1 and B5) are only found in Japan and the western circumpolar Arctic (Alaska, Canada, and Greenland)[67,68]. Based on the age of the non-recombining ancient samples of genotype B in our dataset, the recombination event with genotype C may have occurred after 1.8 kya. This observation also highlights a discrepancy between the modern distribution of subgenotypes B1 and B5 (Supplementary Fig. S9) and their ancient distribution, hinting at a replacement of non-recombinant genotype B (B1 and B5) by the recombinant genotypes B (B2–B4) across most parts of Eastern Eurasia. This replacement may have been facilitated by the recombination event between genotype B and C, which might have conferred advantageous biological properties to the recombinant genotypes. While previous studies have indicated that recombinant genotypes B2–B4 tend to lead to more serious forms of HBV infection, including cirrhosis and development of Hepatocellular carcinoma (HCC), when compared to non-recombinant genotypes B1 and B5[69–73], further functional studies comparing non-recombinant and recombinant genotypes[16] will be needed to understand the mechanisms that caused the replacement in Eastern Eurasia. In the future, it will be possible to compare ancient HBV sequences of genotype B with modern sequences, focusing on the nonsynonymous mutations within these sequences. Furthermore, the sampling of individuals from post 1.8 ka and the detection of the recombinant genotype B in ancient samples could provide clues to the timing of this replacement event.

When assessing the geographical distribution of HBV between ancient and modern times, we observe broad consistency at the genotype level, yet notable variations at the subgenotype level. HBV genotype I can be regarded as a triple recombinant, containing elements from genotypes A, G, and C[74] and has only been found in northwestern China, eastern India, Laos, and Vietnam[12,20,21]. Interestingly, modern distributions indicate no overlap between genotypes I and G, with genotype G predominantly found in many European countries and America. This aligns with the hypothesis that genotype I might have been introduced during the colonial history in the modern age[75]. However, in our recombination analysis, genotype I is modeled as a recombinant of genotypes A and C. Modern genotypes A and C are distributed across Eurasia and North America. Furthermore, ancient genotypes A and C are found in China. These results offer an alternative explanation for the emergence of genotype I.

In summary, our study underscores the necessity of incorporating ancient genomes in the study of HBV's evolutionary history. These ancient sequences reveal a high diversity of HBV in Eastern Eurasia in the past, hinting at this region as a potential geographical origin for genotypes B and D. Our comprehensive analyses, which merge ancient HBV genomes with human DNA and draw upon the archeological context of HBV-infected individuals, emphasize the profound influence of human migration and communication on the dispersal of HBV in ancient times. Furthermore, these analyses shed light on the role of human mobility in driving the evolution of HBV by creating opportunities for recombination events, underscoring the complex interplay between viruses and human populations over millennia.

## Methods

### DNA extraction and library preparation
This study relies on archeological remains previously excavated and incorporates neither new excavation endeavors nor research involving living human or animal subjects. Every newly reported ancient sample

in this study has permission for analysis from custodians of the samples who are co-authors and who affirm that ancient DNA analysis of these samples is appropriate.

Ancient DNA work was carried out in dedicated cleanroom laboratory facilities at the ancient DNA laboratories of Jilin University in Changchun. During sequencing, none of the co-sequenced samples were HBV-related. Moreover, lab personnel were HBV-free. The facility is isolated from contemporary HBV labs, eliminating the risk of modern HBV contamination in our samples. Teeth (https://www.protocols.io/view/tooth-sampling-from-the-inner-pulp-chamber-for-anc-5qpvo5rj9l4o/v2) and pars petrosa (https://www.protocols.io/view/minimally-invasive-sampling-of-pars-petrosa-os-tem-j8nlkem76l5r/v2) were drilled and powder was collected. A total of 50 mg of tooth or pars petrosa powder was used for extraction following the established protocol described in (https://doi.org/10.17504/protocols.io.baksicwe), with the exception that in step 10 the temperature was changed to 50 °C. The extracted DNA was transformed in double-stranded genetic libraries with the use of full, partial, or no uracil DNA-glycosylase (UDG) treatment[40] (https://www.protocols.io/view/non-udg-treated-double-stranded-ancient-dna-librar-3byl47jmzlo5/v1) (https://www.protocols.io/view/full-udg-treated-double-stranded-ancient-dna-libra-5qpvoyq2zg4o/v1)(data S1). Genetic libraries were indexed and amplified before shotgun sequencing. In addition, negative controls were taken along with initial library preparation. These libraries were shotgun sequenced on an Illumina HiSeq X10 or HiSeq 4000 instrument using 2× 150-base-pair (bp) chemistry.

## Screening with MALT
Before performing aligning and taxonomic binning of the obtained reads from the 869 samples with MALT[76] (v.0.5.3), each sample was mapped to the human reference genome (hs37d5) first, using EAGER1[77]. Sequencing quality for each sample was evaluated with FastQC (http://www.bioinformatics.babraham.ac.uk/projects/fastqc/), adapters were clipped and reads were merged using the AdapterRemoval[78] (v.2.2.0) with the --minlength 30 and --minquality 20 options. Merged reads were mapped to the human reference genome using bwa[77] (aln -n 0.01 -l 32). Then the reads that do not map to human are extracted from the bam files, using samtools[79] (v.1.3) (samtools view -f 4). Finally, we used bedtools bamtofastq (v.2.25.0) to convert the bam file to fastq file[80]. These non-human reads were taxonomically assigned by MALT with two different reference datasets: one containing known modern HBV diversity as well as other orthohepadnaviruses[28] and a second database containing parts of modern HBV diversity and other bacteria and virus genomes (see supplement). Both runs used 'semi-global' alignment and a minimum percent identity of 90. For samples that had reads mapped to HBV in the MALT analysis, we used reference sequences (see Section 1) comprising multiple HBV genotypes for comparison using bwa, so as to once again count the reads belonging to HBV in the sample metagenome.

## Enrichment experiment
After screening, those libraries identified as positive for HBV were enriched for HBV DNA using an in-solution target enrichment of HBV following the strategy used in previous ancient HBV work[38,39]. The HBV probes were designed by iGeneTech Co. Ltd (Kit name: AI-HBV-Cap Enrichment Kit, article number: AIHBC), and the experiment was conducted following the manufacturer's instructions. Since the Jiangjialiang site, where sample 95JJLM51 was located, has HBV-positive individuals, and 95JJLM51 yielded a single read aligned to HBV using MALT (despite showing no reads mapped to HBV with bwa), we decided to include it in the enrichment experiment. For some of the individuals (98JJLM9, 95JJLM34), two libraries were built and these two libraries for the same individual were combined when we do the

enrichment: 27 of these were prepared from teeth, while three were prepared from the petrous bone.

## Genotype
To identify the genotype of these individuals, we did a competitive mapping with a combined reference with the EAGER pipeline[77] (see Section 1). AdapterRemoval[78] was used with its default settings to remove adapters from all sequences and reads shorter than 30 bp were discarded. Reads were aligned against the combined reference of the ten hepatitis B genotypes and four NHP strains (see Section 1) using BWA[81] (aln -n 0.01 -l 32) (v.0.7.12) with the same parameters described above. The duplicates were removed by the DeDup module in EAGER[77]. Finally, we count the reads map to each sequence to determine which is the most likely genotype for each of the samples.

For ancient sequences of genotype B with high coverage, we calculated the sequence identity to modern sequences of subgenotype B. For this we computed the number of insertions, deletions, and mismatches between modern and ancient sequences normalized by the total length of the sequence. Missing data in the ancient sequences were not included in the calculations.

## Damage
After determining the genotype of each individual, we choose a reference[82] (see Section 1) and repeat the steps of mapping as described above. To check for the presence of damage patterns characteristic of ancient DNA, consisting of the accumulation of C > T changes due to C deamination at the 5'end of the fragments[83], we use mapDamage v.2.0.9-dirty[84] with default parameters. With exception of the individuals with a few HBV reads in shotgun data and those where full-UDG treatment of the libraries was performed for the in-solution capture experiment, all the others show the typical damage patterns of ancient DNA in the reads mapping to the HBV genome.

## HBV genome reconstruction
After determining the genotype of each individual, we choose a reference[82] (see Section 1) and repeat the steps of mapping as described above. SNP and INDEL calling was carried out with Genome Analysis Toolkit (GATK)[85] UnifiedGenotyped version 3.5 using a quality score of ≥30 and the "EMIT_ALL_SITES" output mode. Then consensus sequences are created using GenConS, which is available in the TOPAS package (-major_allele_coverage 3, -consensus_ratio 0.9, -punishment_ratio 0.8) (https://github.com/subwaystation/TOPAS)[86]. After reconstructing the ancient HBV genomes, we employed previously published methods to evaluate the occurrence of mixed HBV infections in certain individuals[28]. Compared to normal individuals, those with mixed infections have a higher proportion of mixed sites. We assessed signals suggestive of heterozygosity throughout the genome and insertion events at the 5' end of the C gene[28]. The frequencies of the major and minor mutations at each site are calculated and mixed sites are covered at least 10 times, with the major mutation frequency being less than 90%, and the minor mutation frequency greater than 10%. Mixed sites with a major mutation of G and a minor mutation of A, or a major mutation of C and a minor mutation of T, are excluded to ensure that the heterozygosity is not due to ancient DNA damage. Following these criteria, the number of mixed sites is counted, and the overall proportion of positions covered more than 10 times in the dataset that are detected as mixed is calculated. This value serves as the baseline for determining whether an infection is mixed. Previously, no studies had been conducted to separate the sequences of major and minor strain from mixed infection data simultaneously. Consistent with the methods used in previous ancient HBV studies, mixed sites are filtered during the construction of the consensus sequence, retaining only those sites with a frequency greater than 90%. This ensures that the consensus sequences we generate belong to the primary strain.

## Dating of ancient samples

Dating work was carried out in the C-14 laboratory of the Center for Scientific Archeology, Institute of Archeology, Social Sciences of Chinese Academy. Only 13 out of 34 positive individuals have sample dates determined by $^{14}$C dating, using the same samples from which DNA was extracted. The $^{14}$C dates were calibrated using OxCal[87] v.4.4 using the IntCal20 atmospheric curve[88]. Supplementary Table S1 shows the $^{14}$C age and standard deviation for each sample. This is followed by the median probability calibrated years before the present (cal yBP).

Since the individuals from the same site share the same background information, the dates for MY17, MY19, XHM12, XHM16, XHM23, XHM31, NYM9, AT7, AT19, AT24, XBQM20, XBQM47, XBQM86, FLTM18, FLTM97 have been estimated based on the dates of other individuals from that site[34]. 91KLH18 has been dated before[35].

## Initial maximum likelihood phylogenies

An initial maximum likelihood tree was generated using 25 ancient HBV genomes together with modern HBV sequences, and NHP (non-human primates) sequences (see Supplementary data S3 Alignment results). Ancient HBV sequences with at least 50% coverage and a mean coverage greater than 5x were used to compute the maximum likelihood tree. Before the ML tree reconstruction, all the sequences were aligned in MAFFT[89] (v7.305b) (For the reason of low coverage, we exclude XHM16 from the alignment). The resulting alignment was inspected using BioEdit[90] (v.7.2.5) and corrected around large indels when necessary. Using Gblocks, we removed the unresolved positions present in more than 50% of the sequences[91]. An additional stretch of 9 nucleotides (pos. 2990–2998) was masked due to problematic alignment as described as suggested in the previous study (Supplementary Fig. S10)[28]. The maximum likelihood tree was constructed using RAxML[92] (v.8.2.12). We used a GTRCAT substitution model and the rapid bootstrap algorithm with 1000 bootstraps (Supplementary Fig. S3). As nine individuals had mixed HBV infections, we constructed the ML tree, using two datasets with or without the mixed infections. We also constructed a network with the software SplitsTree (v.4.19.2)[93], creating a NeighborNet with uncorrected P distances, using the dataset with the mixed HBV infections.

## Temporal signal assessment and phylogenetic analysis

Root-to-tip regressions were performed to check for a temporal signal in the data using TempEst[44] (v.1.5.3). We used the dataset that included the mixed HBV infections to perform the Temporal signal assessment. The root-to-distances exhibited a strong temporal structure (Supplementary Fig. S4). To perform a time-calibrated phylogenetic analysis, radiocarbon dates for the ancient HBV genomes were used as calibration point in the BEAST analysis[43] (v.2.6.6). To select the appropriate prior model, we conducted path sampling to compare coalescent exponential population, coalescent bayesian skyline, coalescent constant population and birth death skyline tree priors, each of which were combined with either a strict or a relaxed lognormal clock model, using the dataset including the mixed HBV infections. For each model, we executed path sampling with 100 steps of 5 M MCMC iterations and 50% burn-in. We then used the resulting estimates of marginal likelihood to evaluate and compare the performance of each model. Model comparisons supported a relaxed log-normal molecular clock model coupled with a coalescent exponential population prior. After we selected the appropriate prior model, we performed a time calibrated phylogenetic analysis using two datasets with or without the mixed infections. The molecular clock was calibrated using tip dates. For the modern sequences, the dates were set as 0. For the ancient sequences, we used the midrange of $^{14}$C dating or archeological dating as its dates. We used the Gamma distribution site model, GTR substitution model, and relaxed log-normal molecular clocks were tested with coalescent exponential population priors. A uniform distribution between $10^{-9}$ and $10^{-3}$ substitution per site per year was used as a prior for the mean clock rate, based on the range of previous estimates[28,30]. The total Markov chain length was set to 500 M. Then we generate maximum clade credibility (MCC) tree using TreeAnnotator[43] v2.6.2 with the first 10% burn-in[94]. All the parameters have a higher ESS value than 200.

## Recombination analysis

The recombination detection program version 5[46] (RDP5) was used to search for evidence of recombination within the 25 ancient sequences, a selection of 134 modern HBV sequences and non-human primate sequences, and 123 published ancient HBV sequences (Supplementary data S3). Seven recombination methods (RDP, GENECONV, BootScan, MaxChi, Chimaera, SiScan, and 3Seq) were used to detect the recombination event with default parameters. In this analysis, RDP5 constructed maximum likelihood trees for each recombination event separately, using different regions from the presumed major and minor parents in the recombinant. The authenticity of recombination events was confirmed by comparing the position of the recombinant in these two ML trees. For samples of genotype B with low coverage, we performed recombination analysis using SimPlot[48].

## Human population genomic analysis

Only samples with more than 10k SNPs covered in the "1240k-Illumina" panel were involved in downstream human population genomic analysis. We compared the genome sequences of our HBV positive individuals with previously published ancient data[35,42,49,62,95] to the set of genotype panels based on the Affymetrix Axiom Genome-wide Human Origins 1 array (HumanOrigins; 593,124 autosomal SNPs)[96–98]. We grouped the ancient individuals based on archeological culture and genotype of HBV. We carried out Principal Components Analysis (PCA) in the smartpca program of EIGENSOFT[99], using default parameters, the lsqproject: YES[100] and shrinkmode: YES[101]. For ADMIXTURE[102] v.1.3.0, we removed genetic markers with minor allele frequency lower than 1% and pruned for linkage disequilibrium using the-indep-pairwise 200 25 0.2 option[42] in PLINK[103] (version 1.90).

## Reporting summary

Further information on research design is available in the Nature Portfolio Reporting Summary linked to this article.

## Data availability

The raw sequence data reported in this paper have been deposited in the Genome Sequence Archive[105] in National Genomics Data Center[106], China National Center for Bioinformation / Beijing Institute of Genomics, Chinese Academy of Sciences (GSA: CRA013222) that are publicly accessible at https://ngdc.cncb.ac.cn/bioproject/browse/PRJCA020853. The information of published data we used in this study was in data S7.

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

## Acknowledgements
We would like to thank Northwest University, Liaoning University, Institute of Archeology Mongolian Academy of Sciences, Sun Yat-sen University, Institute of Archeology of Russian Academy of Sciences, Inner Mongolia Institute of Cultural Relics and Archeology, Xinjiang Institute of Cultural Relics, Shanxi Provincial Institute of Archeology, Heilongjiang Provincial Institute of Cultural Relics and Archeology, and Zhengzhou University, for sampling permissions. This work was supported by the Natural Science Foundation of China (Grant No. 42372017 and 42072018) Y.Q., the Fundamental Research Funds for the Central Universities (Grant No. 2022CXTD24) Y.Q., National Key Research and Development Project of China (Grant: 2022YFE0203800) J.M., National Social Science Foundation of China, (Grant No, 18CKG026) X.X.

## Author contributions
Y.C., A.H. and J.K. conceived and supervised the study. B.S., S.G., C.L., S.F., F.Z., P.M., X.Y., Y.Q., performed research. Q.Z., J.M., S.C., X.X., D.S., F.L., Al.K., C.H., L.W., W.L., Y.Z., H.Z. provided archeological information and archeological materials. B.S., S.F., X.Y., Y.Q. performed the laboratory work. X.C. performed the AMS dating. B.S. performed the analyses with the support of A.A.V., Ar.K., F.Z. B.S., A.A.V., A. H., J.K. and Y.C. wrote the manuscript with contributions from all authors.

## Competing interests
The authors declare no competing interests.
