## [Peer Review File · Nature Communications]

Origin and dispersal history of Hepatitis B virus in Eastern EurasiaREVIEWER COMMENTS

Reviewer #1 (Remarks to the Author):

NCOMMS-23-44058

Origin and dispersal history of the hepatitis B virus in Eastern Eurasia

Sun et al describe the sequencing and analysis of 30 partial or full ancient HBV genomes from eastern Eurasia. They discuss the diversity of HBV types found at one of the sites, and use their directly dated ancient genomes to expand our understanding of when and where modern recombinant genotypes may have arisen. This is an important insight into the evolution of a pathogen which has a very high morbidity and mortality burden. However, I believe some extra analysis is required before this will be suitable for publication, particularly concerning recombination.

I also note that the authors have not shared any accession numbers for their data so I cannot inspect the genomes.

Abstract

Please describe in the abstract how many of the 30 genomes are full and how many partial.

Line 32-33: A high level of HBV diversity relative to what?

Line 36: I am unsure if the authors really have enough data to conclude that there has been a shift from non-recombinant to recombinant types.

Introduction

Line 49: This doesn't make sense – 60 and 71% add up to more than 100%. Please clarify.

Line 79: Change to "Only two genomes HAVE BEEN recovered from eastern..."

Results

This section is written quite like a discussion (interpretation beyond the data) in many places.

Line 137: Change to "from the" instead of "form the"

Line 148-169: The authors should do an unrooted phylogenetic network (Splitstree or similar) as this will visualise the recombinant nature of data effectively. I am surprised that this wasn't done initially.

Line 192: Change to "we checked" instead of "we check"

Discussion

Do any of the remains have skeletal pathology?

Line 259: "Usually high diversity" – high relative to what? Please contextualise this conclusion.

Line 287-288: the final clause of this sentence does not make sense to me.

Line 289: I think the authors are over-interpreting their data. How can the authors be sure the recombination even did not take place hundreds or thousands of years earlier and only slowly spread to their sampling region? Please back this statement up or change the language used.

Figure 1A

Please show a box on figure 1A that marks out the zoomed in areas shown in B.

Linguistic notes

As far as I know, it is NOT correct for there to be a definite article in front of hepatitis B virus. For example, the title "Origin and dispersal history of the hepatitis B virus in Eastern Eurasia" should be "Origin and dispersal history of hepatitis B virus in Eastern Eurasia". This is true in multiple places in the manuscript, eg the first word of the abstract is an unnecessary definite article.

The authors use a lot of adjectives and adverbs such as "impressive" and "notably" (lines 105-109 but also elsewhere). In the results, it should be for the reader to decide whether these results ARE impressive or notable, not the authors.

The authors flick back and forth between past and present tense to describe work done – please be consistent.

Reviewer #2 (Remarks to the Author):

The authors present data on ancient HBV sequences from Eastern Eurasia. Since there is very little data of ancient HBV sequences from that geographic region, these sequences provide valuable additional information on the distribution of HBV in the past. However, I have concerns about the presentation and interpretation of these data, as outlined in the pdf document that is attached.

Review for Origin and dispersal history of the Hepatitis B virus in Eastern Eurasia

The authors present data on ancient HBV sequences from Eastern Eurasia. Since there is very little data of ancient HBV sequences from that geographic region, these sequences provide valuable additional information on the distribution of HBV in the past. However, I have concerns about the presentation and interpretation of these data, as outlined below.

Main points

Quality of data and analyses

Sequencing

Data S1 suggests that one of the samples (95JJLM51) had no reads mapping to HBV prior to capture. Why is this sample included in the count of 30 here, and why was it captured since there was no indication of it being HBV positive from initial sequencing? In two samples (XBQM86, XHM31) there are relatively less HBV reads to total reads after capture than in the non-captured dataset. In particular, capturing of XBQM86 yielded only one read, which is surprising given that the shotgun data yielded 1201 HBV reads. What happened? In the samples with very few reads, how do the authors ensure that the read did not come from index bleeding/contamination/read carry-over?

Genotype and sub-genotype assignment

SI Section 1 gives a list of references used for genotype assignment, damage analysis and genome reconstruction. How were these selected? Wouldn't some of the previously published ancient sequences be more suitable? In multiple locations in the paper, the authors talk about novel sub-branches or sub-lineages (e.g. lines 191-192, 299, 120-121, 138). Since HBV genotypes are formally sub-divided into sub-genotypes based on sequence identity, discussing possible sub-classifications or novel clades in the light of such a sub-genotype assignment would be more inline with the previous literature. Furthermore, the authors label sub-genotypes in figure 3 without specifying how this was done, and in the discussion (L 289-291) the authors write "*This observation also highlights a discrepancy between the modern distribution of subgenotypes B1 and B5 (Fig. S8) and their ancient distribution*" which appears as if they suggest that their ancient genotype B sequences belong to subgenotypes B1 or B5. In the light of these considerations, actual sub-genotype assignment should be performed. Finally, Kocher et al., have found multiple instances of infection of the same individual with multiple genotypes. Have the authors excluded such double infections and if so, how?

Phylogenetic analysis

The authors should describe how the sequences were chosen that were included in the phylogenetic analyses. The current description implies that only modern sequences are included, which is not the case from the figure. Words such as 'Derived' and 'ancestral' used in line 118-120 are not usually words used to describe phylogenetic trees. Furthermore, talking about 'sub-branches' is unclear, as technically, any branch could be a sub-branch of something. The authors should describe how priors for the BEAST analyses were chosen and how they ascertain that they have chosen a well-fitting model. On lines 164-169, the

description of 11KBM13 and KAP002 is presented without context. Saying that the molecular dating analysis 'confirms' the relationship of these two samples is inappropriate, since there is no previous discussion of the previous result that is being confirmed here is listed. Furthermore, the two sequences discussed here are not visible in the tree. Lines 166 - 169 are inappropriate in the result and should be moved to the discussion. The statement on lines 168-169 suggesting that the introduction of the WENBA genotype into the Tarim basin happened more recently than 9157ya based on the split between 11KBM13 and KAP002 3940ya doesn't follow from the evidence provided, since the genotype may have been present earlier but not sampled. On lines 156 - 161 the authors mention recombination events as possible reasons for low support values in the phylogenetic tree. However, performing a recombination analysis on the full dataset and performing phylogenetic analyses for non-recombinant subsets rather than based on the full genome would be a better way to address those concerns. In addition, the authors should provide analyses describing the recombination event between genotype B and D that they invoke to be responsible for the low support values. Does removing that constraint (by subsetting genome positions or removing recombinant sequences) address the problem with low support values?

Presentation of evidence

Unfortunately, multiple components of the supplementary figures are missing or low quality and analyses are not shown:

Figures S1, S3, S5 are missing. This means lines 116-126, 130-147, 153-156 cannot be assessed, but represent an important part of the results.

Data S2 is missing.

Figures S2, S4, S7 are unreadable.

Figure S10 is included in the SI but isn't cited in the text. The legend states '... as described'. State explicitly where this was described in the figure legend.

L 370, 380 and 390 refer to 'Section 2' but Section 2 in the SI does not support the statements that are made on lines 370, 380, 390.

Information about the 869 screened datasets is missing. The references given on line 91 seemingly as references for the datasets screened do not point to papers that describe the samples that were sequenced and screened, but to the previously published papers that present ancient HBV sequences. Those references do not support the point that is being made here and, if available, should be substituted for references that present the samples that were screened in this study. If this is not possible, information on age, archaeological context and geographic location of the 869 individuals that were screened for this study should be presented in this paper.

Figure 1: The figure is hard to read, the colours and shapes of the different ancient HBV genotypes are difficult to distinguish.

Figure 2: Support values are unreadable even though they are referred to in the text. The description of the x-axis of the tree is wrong, it simply represents time. Details of the blue horizontal bars should be given. The WENBA clade / sequence should be shown.

Figure 3 and Figure S6: The k for these two figures are different (8 in fig 3 vs 9 in fig S6) even though the legend in fig S6 only gives 8 categories. For figure 3, specify how the

subgenotype assignment was done, specify what the “1240k-Illumina” dataset is (provide reference), label the bars in Figure 3b with the sample names.

Paragraph on lines 190-201 is unclear. It's unclear when the authors are talking about the HBV genome and when they're talking about the host genome. 'Heterozygous' is usually not a word used in the context of viral genomes. DA45 is not visible in the tree in figure 2. If the authors want to make a statement about the sub-branch, a tree with that sequence should be shown.

Results of recombination analyses of genotype B sequences should be shown, rather than just stating that no recombination event was detected. For the lower coverage samples, do the sequences have coverage in the appropriate regions to allow the detection of the recombination event? Given that of the 19 genotype B sequences at least 6 have <60% genome coverage at 3x coverage, do the authors trust their negative results for the lower coverage samples? Is the statement on line 126 of most modern HBV being recombinants true based on genotypes or based on prevalence estimated from absolute number of sequences? The authors should provide an adequate reference for this statement. Why were the recombination analyses described on L 126ff focused on recombination of genotype B and C? Appropriate references should be given. Analysis of recombination for the WENBA sequence needs to be described in the results, not just in the discussion.

Dating of samples: On lines 96-98 the authors write “*Radiocarbon dating of these positive individuals places them within the timeframe of 4,130 and 715 calibrated years Before Present (cal BP, Table. S1).*”, creating the impression that all 30 positive samples were radiocarbon dated. However, looking at Table S1 shows that only 13 samples were radiocarbon dated. This should be made explicit in the text.

The missing information described above makes it impossible to adequately judge the full conclusions of this paper.

Interpretation

The paper makes claims and suggestions that are not supported by the data.

The paper reports ‘30 ancient HBV genomes’ in multiple places (L 30, 203 etc), even though some of them had almost no coverage. The authors should state in those places that most of the genomes were partial, how many genomes actually had enough coverage for the inclusion in phylogenetic analyses and what criteria they used to determine this.

Statements on origin, past distribution and human migration

The paper suggests to have indications of the origin of genotypes in specific geographic regions or makes inferences from the past distribution of genotypes. E.g lines 84ff “*The newly reconstructed ancient HBV genomes suggest Eastern Eurasia as a plausible origin for genotypes B, C, and D.*”, “*These ancient HBV genomes unveil a rich diversity of genotype B in Eastern Eurasia, suggesting a potential origin of genotype B within this region.*” (L 231-233), the suggestion on L 213-215 that the distribution of 7 genotype B sequences 3000-1600 yBP agrees with the modern distribution, the co-existence of WENBA and genotypes A and D (L 210-212), the “reappearance of genotype D” (L 223-224), and suggestions about the origin of genotype C (L 244-245), genotype D (L281). Likewise,

diversity of sequences in an area is not indicative of an origin of a genotype in that area. Given the massive spatial and temporal undersampling of HBV, such claims should not be made or should be carefully discussed in light of undersampling and possible sampling biases.

While an effect of human migration on the distribution of HBV genotypes and subgenotypes can be expected, the discussions in this paper relating the geographic distribution of HBV genotypes to human migrations need to be more nuanced and presented in the wider context.

For example, on line 221-223 the authors state that “*The close relationship observed in the phylogenetic tree between BRE008, DA27, SHK001, DA222, and MAY017 with our genotype D individuals is consistent with the cultural interactions of these ancient societies.*” A reader not familiar with those samples cannot assess this statement at all without knowing where those samples are from, how old they are, and which culture those individuals belong to.

Statements on recombination

The analyses and discussion of recombination in this paper are insufficient.

On lines 283 - 286 the authors suggest “*Our findings provide evidence of recombination events occurring between genotypes B and C, resulting in the emergence of subgenotypes B2-B4*”. The authors do not show this, in fact they do not show the results of the recombination analyses that they did for genotype B where they mention that they did not find recombination events at all. The observation of recombination events between genotypes B and C has been made previously and is well known in the field (e.g. <https://www.ncbi.nlm.nih.gov/pmc/articles/PMC136227/>) and should not be presented as novel here. Furthermore, related to the discussion of the timing of the recombination event (L 288-292), the tree in fig 2 suggests at least two recombination events, since subgenotype B3 falls basal and subgenotypes B2 and B4 fall within the clade of non-recombinant sequences. The date of the recombination event therefore cannot be correct based on the tree structure. Speculation about fitness advantages of recombinant subgenotypes seems premature, since subgenotype replacement could also have been favoured by factors not directly linked to viral fitness, such as population replacement for reasons not related to HBV. Suggestions for experimental work to elucidate fitness differences should be more concrete than suggested on line 297. On lines 276 - 282 the authors introduce a possible recombination event between genotype A and WENBA. The authors should specify which “Previous analyses” they are referring to by a reference or a figure, and the analysis needs to be described. Figure S7 needs to be discussed in the results. Finally, on lines 310-313, the authors introduce a novel hypothesis about the origin of genotype I. Their hypothesis could be tested by performing additional recombination analyses, and the authors should do this rather than just speculating.

Minor points

L 29: s/precedes/exceeds

L 41: The reference is indirect. If the authors wish to reference a paper giving evidence for the origin of HBV, they should consider referencing studies such as

<https://doi.org/10.1073/pnas.1908072116> or <https://doi.org/10.1371/journal.pbio.1000495>.

L 49-50: The percentage of infections by genotypes B and C in China don't add up to 100%.

L 60-61: Given that the authors suggest that much earlier migrations shaped HBV diversity, it seems strange that they introduce the concept of dissemination of pathogens through globalisation and industrialisation which are processes that happened much later.

L 62 - 63: "Remarkably, HBV only infects humans and a few other primate species²⁶" Given that closely related viruses (that are also classified as HBV) are also infecting rodents, bats and other mammals, this statement is wrong.

L 64: Typo "... tightly linked **to** human..."

L 70: 10,000 isn't really a deep timescale.

L 72-75: Members of the WENBA lineage were already described (although not named as such) in <https://doi.org/10.1038/s41586-018-0097-z> and <https://doi.org/10.7554/eLife.36666.001>.

L 79: Typo "Only two genomes **have** been recovered."

L 96: Reference to 'these positive samples' where it is unclear which previously mentioned sample set is referred to.

L 105: The use of 'impressive' here is inappropriate.

L 106: The use of 'successfully' here is inappropriate.

L 109-111: This statement is out of place in the results and should be moved to the discussion.

L 115: The tree shows only 20 sequences, although the WENBA sequence may have been collapsed?

L 119: what does 'more derived' mean?

L 161: The reference to figure S4 seems inappropriate here.

L 186: This point is repeated from line 179. Also, I'm not an expert in interpreting ADMIXTURE results, but I don't see a consistent trend in differences between the genetic profile for individuals with genotype B and D.

L 190: '(HBV genome ~~have been~~ published in 2018)'

L 201: Reference 31 seems out of place here.

L233-238: It's not clear what the 'In contrast' refers to. The discussion of genotype A origins does not tie in with any data that is presented and seems out of place.

L 234: Stray 'S' at the beginning of the line.

L 246 - 254: This is largely a repeat from the results section.

L 256 and 258: The authors should be explicit with what they mean by 'this region'.

L 382: C->T changes occur at the 5' end, not the 3' end, G->A changes on the 3' end.

L 399: The authors state that 14 out of 28 positive samples were radiocarbon dated, but the paper mentions 30 positive samples? Table S1 only gives radiocarbon dates for 13 samples. Why?

L 404-406: do the dates assigned from other individuals from the same site agree with the context that the non-radiocarbon dated samples were found in?

L 408: The main text says that 21 sequences were included, not 20?

L 410: State which coverage depth was used for determining the 50% coverage.

L 413: What is A336? Give reference for that sample.

L 420: Fig S4 doesn't support the statement that is being made.

L 424: Fig S2 doesn't support the statement that is being made.

L 444: What is the "1,240k panel"? Give reference.

Archaeological background for Bayanbulag gives radiocarbon dates for two samples (AT6, AT8) that aren't listed in the title of the section.

Reviewer #3 (Remarks to the Author):

The study describes sequencing data from 30 ancient HBV genomes dated between approximately 4,130 and 715 years ago from 13 sites in Eastern Eurasia. HBV is a globally distributed pathogen and the history of its infection in humans dates back 10,000 years. The results demonstrated that genotypes B, C and D may have originated in Eastern Asia. Furthermore, a high level of HBV diversity in a single location in Xinjiang, characterized by the presence of three different genotypes (A, B, D), underlines the importance of migrations and human interactions in the spread of HBV. Furthermore, the authors report the identification of a transition from non-recombinant subgenotypes (B1, B5) to recombinant subgenotypes (B2-B4). This suggests a change in epidemiological dynamics in Eastern Eurasia over time. The authors highlight that the study reveals the regional origins of the predominant genotypes and the changes in viral subgenotypes over the centuries.

In my opinion, the study is interesting and presents new ancient HBV sequencing data in samples from Eastern Eurasia. The methodology for obtaining old HBV data appears to be adequate. However, it is necessary to have criteria to validate the viral DNA data obtained from these very old samples. A major challenge for any study of very ancient DNA is the accumulation of post-mortem damage to the genome of interest. These damages include fragmentation, nucleotide deamination, and polymerase-blocking lesions, such as molecular cross-linking, resulting from enzymatic and chemical reactions. Predictably recurring forms of damage, such as the tendency for cytosine deamination to occur more frequently near the 3' and 5' ends of fragmented DNA molecules, also provide a means of addressing contamination issues and inferring the authenticity of a recovered sequence. through statistics. pattern analysis. Therefore, I suggest that the authors present more evidence of the authenticity of the analyzed DNA.

I am also concerned about the sequence dataset used (including old and new HBV DNA sequences) as well as the results of the phylogenetic and phylodynamic analyzes presented. Methods that predict the temporal dynamics and phylogeography of recent virus emergence have been remarkably effective in reconstructing recent virus evolutionary histories. Extrapolating substitution rates to longer periods could provide the means to reconstruct much deeper evolutionary histories of viruses. However, a number of recent developments challenge the applicability of such methods to the widely accepted concepts of virus evolutionary timescales. Noteworthy, HBV is a partially double-stranded DNA virus with a circular genome of just 3,200 nucleotides. It employs error-prone reverse transcriptase (RT) in part of its replication process, making its evolutionary rate (substitution/site/year) closer to that of RNA viruses than DNA viruses. Furthermore, the four open reading frames (ORFs), which encode the surface (S), nucleocapsid (C), and X proteins and the polymerase (P), largely overlap, restricting viral evolution because synonymous mutations in one frame are often non-synonymous in another. Previous studies have already demonstrated a contrast between high replacement rates in the short term and apparently low replacement rates in the long term (rate mismatch). There is also a high diversity of intra-host HBV, creating even more difficulties in the analysis.

In addition, the manuscript presents the classification of ancient HBV sequences using the current taxonomy of genotypes and subgenotypes. A previous scientific paper (cited by the authors) presented the occurrence of a West Eurasian Neolithic to Bronze Age (WENBA) HBV lineage. Why couldn't the same happen in Eastern Eurasia?

In my opinion, all these questions should be answered and included in the manuscript.

Point-by-Point Response to Reviewer Comments for NCOMMS-23-44058A
“Origin and dispersal history of Hepatitis B virus in Eastern Eurasia”

Thank you very much for careful and thorough reading of this manuscript and for the thoughtful comments and constructive suggestions. We truly appreciate all your comments and suggestions, which helped to improve the quality of this manuscript. In response to the three reviewers’ advice, we have made the necessary revisions to the manuscript. Our point-by-point responses to the reviewers' comments are highlighted in blue, and the changes made to the Main Text and Supplement are marked accordingly. Additionally, we have provided a clean version of the manuscript and supplement for the convenience of the reviewers.

Reviewer #1 (Remarks to the Author):

Sun et al describe the sequencing and analysis of 30 partial or full ancient HBV genomes from eastern Eurasia. They discuss the diversity of HBV types found at one of the sites, and use their directly dated ancient genomes to expand our understanding of when and where modern recombinant genotypes may have arisen. This is an important insight into the evolution of a pathogen which has a very high morbidity and mortality burden. However, I believe some extra analysis is required before this will be suitable for publication, particularly concerning recombination.

We sincerely thank the reviewer for their time and attention devoted to reviewing our work. The constructive feedback is greatly appreciated. We have carefully revised the manuscript based on the reviewer’s comments.

Q1: I also note that the authors have not shared any accession numbers for their data so I cannot inspect the genomes.

The raw sequence data reported in this paper have been deposited in the Genome Sequence Archive in National Genomics Data Center, China National Center for Bioinformatics / Beijing Institute of Genomics, Chinese Academy of Sciences (GSA: CRA013222) that are publicly accessible at <https://ngdc.cncb.ac.cn/bioproject/browse/PRJCA020853>.

Abstract

Q2: Please describe in the abstract how many of the 30 genomes are full and how many partial.

We have modified the abstract accordingly. We have provided a more detailed account of the number of genomes covered in full and the quality of other genomes. Please see line 31-32 in clean version.

Q3: Line 32-33: A high level of HBV diversity relative to what?

The diversity is in comparison with Western Eurasia at this time. Published data indicates that two genotypes were present in Western Eurasia between 5000 and 3000 years ago, and at a site in Xinjiang, we discovered the existence of three genotypes, indicating that the diversity at this site in Xinjiang is higher than the

diversity in Western Eurasia. Thus, we suggest that the HBV found in Xinjiang exhibits exceptionally high diversity. We changed the description in the main text add “compared to Western Eurasia” to the end of the original sentence. Please see line 33-34 in clean version.

Q4: Line 36: I am unsure if the authors really have enough data to conclude that there has been a shift from non-recombinant to recombinant types.

It is true that the sample size in our study appears to be small compared to the larger sample sizes common in modern HBV studies, however, ancient DNA studies have their unique characteristics. After conducting recombination analysis, we found that all ancient sequences of genotype B are non-recombinants, while the modern HBV of genotype B prevalent in this region today are recombinants. To ensure precision in our expression, we have changed the wording in abstract.

Introduction

Q5: Line 49: This doesn't make sense – 60 and 71% add up to more than 100%. Please clarify.

Our sentence meant that 60.5% of global genotype B patients and 71% of global genotype C patients are in China. In order to avoid misunderstanding, we have revised this statement to “In China, Genotype B are responsible for 27.9% and genotype C are responsible for 64.4% of HBV infections.” Please see line 50-51 in clean version.

Q6: Line 79: Change to “Only two genomes HAVE BEEN recovered from eastern...”
We have made the modifications accordingly.

Results

Q7: This section is written quite like a discussion (interpretation beyond the data) in many places.

We have relocated the detailed explanations and analyses of the data to the discussion section.

Q8: Line 137: Change to “from the” instead of “form the”

We have made the modifications accordingly.

Q9: Line 148-169: The authors should do an unrooted phylogenetic network (Splitstree or similar) as this will visualise the recombinant nature of data effectively. I am surprised that this wasn't done initially.

We have constructed an unrooted phylogenetic network according to the reviewer's suggestion, which indicates different phylogenetic positioning for some of the recombinant lineages. That means these lineages are from different recombination events. The subgenotypes B2, B3, and B4, which have recombined with genotype C, are located between the non-recombinant lineages of genotype B and genotype C. This result is consistent with the results of our recombination analysis. Please see Fig. S6.

Q10: Line 192: Change to “we checked” instead of “we check”

We have made the modifications accordingly.

Discussion

Q11: Do any of the remains have skeletal pathology?

We cross-checked the bones with paleoanthropologists and paleopathologists. These remains do not have visible pathological lesions. We have added this information to lines 95-96 of the main text.

Q12: Line 259: “Usually high diversity” – high relative to what? Please contextualise this conclusion.

See our answer to point Q3. We've refined this part of the information in the main text. Please see lines 308-310.

Q13: Line 287-288: the final clause of this sentence does not make sense to me.

We have removed that section to make the text more concise. Please see lines 336-340.

Q14: Line 289: I think the authors are over-interpreting their data. How can the authors be sure the recombination even did not take place hundreds or thousands of years earlier and only slowly spread to their sampling region? Please back this statement up or change the language used.

We have changed to that “However, we cannot preclude the possibility that this recombination event occurred in another region thousands of years ago, and subsequently spread to Xinjiang.”

Figure 1A

Q15: Please show a box on figure 1A that marks out the zoomed in areas shown in B.

We have made the modifications accordingly. Please see Fig. 1.

Linguistic notes

Q16: As far as I know, it is NOT correct for there to be a definite article in front of hepatitis B virus. For example, the title “Origin and dispersal history of the hepatitis B virus in Eastern Eurasia” should be “Origin and dispersal history of hepatitis B virus in Eastern Eurasia”. This is true in multiple places in the manuscript, eg the first word of the abstract is an unnecessary definite article.

We have made the modifications accordingly.

Q17: The authors use a lot of adjectives and adverbs such as “impressive” and notably” (lines 105-109 but also elsewhere). In the results, it should be for the reader to decide whether these results ARE impressive or notable, not the authors.

We have made the modifications accordingly.

Q18: The authors flick back and forth between past and present tense to describe work done – please be consistent.

We have made the modifications accordingly.

Reviewer #2 (Remarks to the Author):

The authors present data on ancient HBV sequences from Eastern Eurasia. Since there is very little data of ancient HBV sequences from that geographic region, these sequences provide valuable additional information on the distribution of HBV in the past. However, I have concerns about the presentation and interpretation of these data, as outlined below.

We are writing to express my deep appreciation for the constructive feedback you have provided on my manuscript. The time and effort you dedicated to reviewing my work are immensely valued, and your expert critique has significantly contributed to the refinement of my research. The detailed comments have not only guided us in making substantial improvements to the document but have also deepened my understanding of the subject matter, enhancing the scholarly value of my work. Your rigorous approach has encouraged us to present my findings with greater precision.

Main points

Quality of data and analyses

Sequencing

Q1: Data S1 suggests that one of the samples (95JJLM51) had no reads mapping to HBV prior to capture. Why is this sample included in the count of 30 here, and why was it captured since there was no indication of it being HBV positive from initial sequencing?

This sample had a single read aligned to HBV using MALT but showed no reads mapped to HBV with bwa. Given the presence of HBV-positive individuals at this archaeological site, we included a sample in the analysis (95JJLM51). Please see 99-102 in clean version.

Q2: In two samples (XBQM86, XHM31) there are relatively less HBV reads to total reads after capture than in the non-captured dataset. In particular, capturing of XBQM86 yielded only one read, which is surprising given that the shotgun data yielded 1201 HBV reads. What happened? In the samples with very few reads, how do the authors ensure that the read did not come from index bleeding/contamination/read carry-over?

Thank you very much for your question, we do need to clarify this a bit. The decline in quality of the two samples post-capture was due to issues during the capture process, therefore the capture was not successful, and it is not a result of contamination. Damage patterns can be observed when there is a sufficient number of reads, and there were no modern HBV samples on the same sequencing line at the time of sequencing, plus the individual handling the process was not infected with HBV. Additionally, the laboratory is isolated from modern experimental environments, eliminating the possibility of contemporary HBV contamination.

Genotype and sub-genotype assignment

Q3: SI Section 1 gives a list of references used for genotype assignment, damage analysis and genome reconstruction. How were these selected? Wouldn't some of the previously published ancient sequences be more suitable?

We select these genomes as our reference sequences based on a study that provides a comprehensive set of HBV reference sequences at both the genotype

and subgenotype levels. One reason we decided not to use ancient sequences as reference is that the modern genomes we selected are complete, whereas the ancient sequences available often lack comprehensive coverage. Please see supplement section 1.

Q4: In multiple locations in the paper, the authors talk about novel sub-branches or sub-lineages (e.g. lines 191-192, 299, 120-121, 138). Since HBV genotypes are formally sub-divided into sub-genotypes based on sequence identity, discussing possible sub-classifications or novel clades in the light of such a sub-genotype assignment would be more inline with the previous literature. Furthermore, the authors label sub-genotypes in figure 3 without specifying how this was done, and in the discussion (L 289-291) the authors write “This observation also highlights a discrepancy between the modern distribution of subgenotypes B1 and B5 (Fig. S8) and their ancient distribution” which appears as if they suggest that their ancient genotype B sequences belong to subgenotypes B1 or B5. In the light of these considerations, actual sub-genotype assignment should be performed.

Thank you very much for your suggestion, which has helped to enhance the rigor and completeness of our paper. Prior to this, we had performed genotyping based on sequence consistency but had not displayed it in the results section, which might have caused some confusion. In line with your advice, we have now included the results of the genotyping in the supplementary materials, and they are finally presented in Figure 4. The genotyping results shown are based on the phylogenetic tree outcomes, although there are a few samples whose positions on the evolutionary tree differ from the genotyping results. Please see table S2.

Q5: Finally, Kocher et al., have found multiple instances of infection of the same individual with multiple genotypes. Have the authors excluded such double infections and if so, how?

Thank you very much for your question, which served as a valuable reminder. We have used the method described in Kocher's paper to test for individuals with mixed infections which allowed us to identify eight cases of mixed infection. We have supplemented our phylogenetic analyses by a new one in which we excluded the genomes recovered from these individuals. The results indicate that after the removal of mixed infections, the topological structure of genotype B has changed. Please see Fig. S3b and Fig. S5a. Here is the description of the method in the original article.

“To assess the presence of mixed HBV infections in some individuals, we looked for heterozygous-like signals along the HBV genome sequence in sequencing read alignments. Our rationale was that in case of a mixed infection, one should observe more frequent heterozygous-like positions than expected with molecular and sequencing artifacts only. For each sample, we computed the relative support of the two major variants at each HBV genomic position, for which we also derived a 95% confidence interval using Wilson’s method (as implemented in the R package binom). We then defined heterozygous-like positions as positions being covered at least 10x and for which the support of both major and second major variants was significantly below 90% and above 10% respectively. Cases where the major variant was C or G and the second variant was T or A, respectively, were excluded, in order to discard any heterozygous like signal possibly due to DNA damage. The baseline probability of a position to be detected as heterozygous-like was estimated as the overall fraction of heterozygous-like positions over positions covered >10x in the complete dataset. A given sample was considered as an outlier and indicative of a mixed infection if it contained a number of heterozygous-like positions that exceeded the 95% quantile of a binomial distribution parameterized with the above-mentioned baseline probability and the number of positions covered >10x in the given sample as number of trials. The results were identical when using the 99% quantile instead.”

Phylogenetic analysis

Q6: The authors should describe how the sequences were chosen that were included in the phylogenetic analyses. The current description implies that only modern sequences are included, which is not the case from the figure.

We have made the modifications. We used the newly reconstructed ancient genomes that have over 50% genome coverage and a mean coverage greater than 5x. Please see lines 124-127 in clean version.

Q7: Words such as ‘Derived’ and ‘ancestral’ used in line 118-120 are not usually words used to describe phylogenetic trees. Furthermore, talking about ‘sub-branches’ is unclear, as technically, any branch could be a sub-branch of something.

We changed the sequence to that “The genome of XBQM86, recovered from the Quanergou site, represents the second deepest branch in the lineage leading to genotype A”. Please see lines 131-133 in clean version.

Q8: The authors should describe how priors for the BEAST analyses were chosen and how they ascertain that they have chosen a well-fitting model.

We have done this analysis and found that it is not advisable to directly apply the prior models from previous studies, as the prior model most suitable for our dataset differs from those published. Thank you very much for your proposal; it is of great scientific significance to our research. Please see lines 171-174 and Table. S3.

Q9: On lines 164-169, the description of 11KBM13 and KAP002 is presented without context. Saying that the molecular dating analysis ‘confirms’ the relationship of these two samples is inappropriate, since there is no previous discussion of the previous result that is being confirmed here is listed. Furthermore, the two sequences discussed here are not visible in the tree. Lines 166 - 169 are inappropriate in the result and should be moved to the discussion. The statement on lines 168-169 suggesting that the introduction of the WENBA genotype into the Tarim basin happened more recently than 9157ya based on the split between 11KBM13 and KAP002 3940ya doesn’t follow from the evidence provided, since the genotype may have been present earlier but not sampled.

We have adjusted the presentation of the phylogenetic tree in Figure 3, which now clearly displays sequences of WENBA, such as 11KBM13 and KAP002. Additionally, we have moved the discussion-like statements to the discussion section. In the main text, we changed it into that “we cannot exclude the possibility that there may exist samples older than 11KBM13 in the region, which could potentially reflect different transmission patterns of WENBA.” Please see lines 192-194 and 293-299.

Q10: On lines 156 - 161 the authors mention recombination events as possible reasons for low support values in the phylogenetic tree. However, performing a recombination analysis on the full dataset and performing phylogenetic analyses for non-recombinant subsets rather than based on the full genome would be a better way to address those concerns. In addition, the authors should provide analyses describing the recombination event between genotype B and D that they invoke to be responsible for the low support values. Does removing that constraint (by subsetting genome positions or removing recombinant sequences) address the problem with low support values?

We have conducted recombination analysis using all the samples utilized for phylogenetic analysis, and the results indicate that, with the exception of subgenotype B1, B5, and our ancient sequences, all other B-type samples are recombinants. Additionally, we observed that individuals with genotype D can

commonly be modeled as recombinants of WENBA and A. The analysis results are displayed in Fig. S7, data S4, and data S5. We constructed an unrooted tree with all the samples, and in this result, we noted that the topology of the recombinant individuals differs from that in the maximum likelihood tree and the MCC tree. Please see Fig. S3 and Fig. S5 and Fig. S6 in supplement.

Presentation of evidence

Q11: Unfortunately, multiple components of the supplementary figures are missing or low quality and analyses are not shown: Figures S1, S3, S5 are missing. This means lines 116-126, 130-147, 153-156 cannot be assessed, but represent an important part of the results. Data S2 is missing. Figures S2, S4, S7 are unreadable.

We will upload the clearest file, so that all attached files are clear and readable, please check the attached PDF file of supplement.

Q12: Figure S10 is included in the SI but isn't cited in the text. The legend states '... as described'. State explicitly where this was described in the figure legend.

We have cited Fig. S10 in the text. Please see line 466-468.

Q13: L 370, 380 and 390 refer to 'Section 2' but Section 2 in the SI does not support the statements that are made on lines 370, 380, 390.

We are sorry for this mistake and we made an error in the Section that was cited. We have made the correction.

Q14: Information about the 869 screened datasets is missing. The references given on line 91 seemingly as references for the datasets screened do not point to papers that describe the samples that were sequenced and screened, but to the previously published papers that present ancient HBV sequences. Those references do not support the point that is being made here and, if available, should be substituted for references that present the samples that were screened in this study. If this is not possible, information on age, archaeological context and geographic location of the 869 individuals that were screened for this study should be presented in this paper.

We have added the site information for all 869 individuals that we screened in the supplement, including those sites where no HBV positive individuals were detected. Please see supplement Section 2.

Q15: Figure 1: The figure is hard to read, the colours and shapes of the different ancient HBV genotypes are difficult to distinguish.

We have implemented new colors to represent different genotypes and used different maps as backgrounds, which significantly improved the distinguishability of the individuals. Please see Fig. 1 and Fig. 2.

Q16: Figure 2: Support values are unreadable even though they are referred to in the text. The description of the x-axis of the tree is wrong, it simply represents time. Details of the blue horizontal bars should be given. The WENBA clade / sequence should be shown.

We have added a phylogenetic tree to the supplement that illustrates the 95% confidence interval of the MCRA represented by blue horizontal bars, and in the main text's figure, we have shown the branch of WENBA where our ancient individuals are situated.

Q17: Figure 3 and Figure S6: The k for these two figures are different (8 in fig 3 vs 9 in fig S6) even though the legend in fig S6 only gives 8 categories.

We have changed all of it into 9.

Q18: For figure 3, specify how the subgenotype assignment was done, specify what the “1240k-Illumina” dataset is (provide reference), label the bars in Figure 3b with the sample names.

As suggested, we have now included the results of the sequence identity analysis in Table S2 and have added a reference annotation to the "1240k-Illumina" dataset. Additionally, we have used the names of all samples as labels in Fig. 4b. Please see table S2, and Fig. 4, and line 527 in clean version.

Q19: Paragraph on lines 190-201 is unclear. It's unclear when the authors are talking about the HBV genome and when they're talking about the host genome.

We have attached the figures corresponding to each described sentence right after the sentences, which should better assist readers in understanding the parts being discussed. Please see lines 212-229.

Q20: ‘Heterozygous’ is usually not a word used in the context of viral genomes.

We have changed "Heterozygous" to "being mixed" to describe the presence of different types of bases at the same position. Please see lines 237-240.

Q21: DA45 is not visible in the tree in figure 2. If the authors want to make a statement about the sub-branch, a tree with that sequence should be shown.

We have reconstructed the maximum likelihood tree including the sample DA45. Please see Fig. S3c.

Q22: Results of recombination analyses of genotype B sequences should be shown, rather than just stating that no recombination event was detected. For the lower coverage samples, do the sequences have coverage in the appropriate regions to allow the detection of the recombination event? Given that of the 19 genotype B sequences at least 6 have <60% genome coverage at 3x coverage, do the authors trust their negative results for the lower coverage samples?

Thank you very much for your suggestions. We have presented the results of our recombination analysis in the supplement. Please see Fig. S7. For samples with low coverage, we performed recombination analysis using SimPlot, which also did not detect any recombination events with genotype C, consistent with the results from our RDP5 analysis of all samples. Since the coverage for HBV of genotype B post-dating 1800 is over 60%, we have made the most objective assessment possible in our conclusions. We did not detect recombination events in ancient HBV of genotype B from around 1000 years ago. Due to their lower quality, this does not definitively indicate the absence of recombination. Samples predating 1800, as well as even older samples, have genome coverages greater than 80%, leading us to believe that the lower quality of some samples did not affect our analysis and judgment regarding recombination events. Of course, higher quality data would aid in increasing the accuracy of the recombination analysis for genotype B from around 1000 years ago.

Q23: Is the statement on line 126 of most modern HBV being recombinants true based on genotypes or based on prevalence estimated from absolute number of sequences? The authors should provide an adequate reference for this statement. Why were the recombination analyses described on L 126ff focused on recombination of genotype B and C?

When we mention that the majority of genotypes are recombinants, it is in reference to the absolute number of sequences. Our recombination analysis did not particularly emphasize the recombination between B and C, as the

recombination events within genotype B primarily involve recombination with genotype C.

Q24: Appropriate references should be given. Analysis of recombination for the WENBA sequence needs to be described in the results, not just in the discussion.

We have included a separate discussion on the recombination analysis in the results section. Please see lines 196-208.

Q25: Dating of samples: On lines 96-98 the authors write “Radiocarbon dating of these positive individuals places them within the timeframe of 4,130 and 715 calibrated years Before Present (cal BP, Table. S1).”, creating the impression that all 30 positive samples were radiocarbon dated. However, looking at Table S1 shows that only 13 samples were radiocarbon dated. This should be made explicit in the text.

Thank you very much for your comments, which we have revised and clarified in the main text. Please see lines 102-105.

Interpretation

Q26: The paper makes claims and suggestions that are not supported by the data. The paper reports ‘30 ancient HBV genomes’ in multiple places (L 30, 203 etc), even though some of them had almost no coverage. The authors should state in those places that most of the genomes were partial, how many genomes actually had enough coverage for the inclusion in phylogenetic analyses and what criteria they used to determine this.

We have made the modifications according to your suggestions. We have provided a more detailed account of the number of genomes covered in full and the quality of other genomes. Please see line 31-32 in the clean version of the manuscript. The standards we apply to both modern data and previously published ancient sequences and our sequences are consistent, requiring that more than 50% of the positions are covered at a depth of 3x. Please see lines 124-127.

Statements on origin, past distribution and human migration

Q27: The paper suggests to have indications of the origin of genotypes in specific geographic regions or makes inferences from the past distribution of genotypes. E.g lines 84ff “The newly reconstructed ancient HBV genomes suggest Eastern Eurasia as a plausible origin for genotypes B, C, and D.“, “These ancient HBV genomes unveil a rich diversity of genotype B in Eastern Eurasia, suggesting a potential origin of genotype B within this region.” (L 231-233), the suggestion on L 213-215 that the distribution of 7 genotype B sequences 3000-1600 yBP agrees with the modern distribution, the co-existence of WENBA and genotypes A and D (L 210-212), the “reappearance of genotype D” (L 223-224), and suggestions about the origin of genotype C (L 244-245), genotype D (L281). Likewise, diversity of sequences in an area is not indicative of an origin of a genotype in that area. Given the massive spatial and temporal undersampling of HBV, such claims should not be made or should be carefully discussed in light of undersampling and possible sampling biases.

We appreciate your suggestion. Among the sequences we added in this version of the manuscript, there is a 5,000-year-old genotype B sequence that fall basal to all the modern and ancient sequences of genotype B. Given the diversity of genotype B across the Eurasian continent, we believe there is a strong possibility that it originated in Eurasia. We acknowledge that unsampled sequences may reveal different histories, so we have included a note stating that we do not exclude the possibility of other origins.

Q28: While an effect of human migration on the distribution of HBV genotypes and subgenotypes can be expected, the discussions in this paper relating the geographic

distribution of HBV genotypes to human migrations need to be more nuanced and presented in the wider context. For example, on line 221-223 the authors state that “The close relationship observed in the phylogenetic tree between BRE008, DA27, SHK001, DA222, and MAY017 with our genotype D individuals is consistent with the cultural interactions of these ancient societies.” A reader not familiar with those samples cannot assess this statement at all without knowing where those samples are from, how old they are, and which culture those individuals belong to.

We have introduced the culture each individual belongs to after each individual’s name in the main text. “The close relationship observed in the phylogenetic tree between BRE008 (hun-Xianbei), DA27 (hun-sarmatian), SHK00131, DA222 (karluk), and MAY017 (Golden Horde)” Please see lines 263-266.

Statements on recombination

Q29: The analyses and discussion of recombination in this paper are insufficient. On lines 283 – 286 the authors suggest “Our findings provide evidence of recombination events occurring between genotypes B and C, resulting in the emergence of subgenotypes B2-B4”. The authors do not show this, in fact they do not show the results of the recombination analyses that they did for genotype B where they mention that they did not find recombination events at all. The observation of recombination events between genotypes B and C has been made previously and is well known in the field (e.g. <https://www.ncbi.nlm.nih.gov/pmc/articles/PMC136227/>) and should not be presented as novel here. Furthermore, related to the discussion of the timing of the recombination event (L 288-292), the tree in fig 2 suggests at least two recombination events, since subgenotype B3 falls basal and subgenotypes B2 and B4 fall within the clade of non-recombinant sequences. The date of the recombination event therefore cannot be correct based on the tree structure. Speculation about fitness advantages of recombinant subgenotypes seems premature, since subgenotype replacement could also have been favoured by factors not directly linked to viral fitness, such as population replacement for reasons not related to HBV.

Thank you very much for your recommendation. In the supplement, we have presented evidence of recombination between genotype B and genotype C. In addition, we have shown all results of the recombination analysis, not limited to recombination between genotypes B and C. As the recombination of genotypes B and C is well known, we have made some modifications to this section of the description. Regarding the timing of recombination, based on your suggestion and our analysis, it is confirmed that there have been at least two independent recombination events between B and C, occurring at different times. Our results confirm that neither of these recombination events occurred before 1800 years ago. As for the hypothesis that recombinants may have an adaptive advantage, we also speculate on such a possibility, although further conclusions would require the support of biological experiments. Hence, we have also indicated that there could be migration of populations or other factors that have affected the spread and transmission of HBV, ultimately leading to this replacement event. Please see Fig. S7, and data S4 and data S5. Please see lines 336-348.

Q30: Suggestions for experimental work to elucidate fitness differences should be more concrete than suggested on line 297.

Thank you very much for your suggestion. We have added some suggestions in our main text. Please see lines 355-359.

Q31: On lines 276 – 282 the authors introduce a possible recombination event between genotype A and WENBA. The authors should specify which “Previous analyses” they

are referring to by a reference or a figure, and the analysis needs to be described. Figure S7 needs to be discussed in the results.

We have referred to the reference for the recombination events between A and WENBA, and we have included the results of the recombination analysis in the Results section for a detailed description. Please see lines 196-208.

Q32: Finally, on lines 310-313, the authors introduce a novel hypothesis about the origin of genotype I. Their hypothesis could be tested by performing additional recombination analyses, and the authors should do this rather than just speculating.

Thank you very much for your suggestions. In our recombination analysis, we also observed recombination events involving genotype I, where genotype We could be modeled as a recombinant of genotypes A and C.

Minor points

Q33: L 29: s/predates/exceeds L 41: The reference is indirect. If the authors wish to reference a paper giving evidence for the origin of HBV, they should consider referencing studies such as <https://doi.org/10.1073/pnas.1908072116> or <https://doi.org/10.1371/journal.pbio.1000495>.

We have made the modifications according to your suggestions.

Q34: L 49-50: The percentage of infections by genotypes B and C in China don't add up to 100%.

Please see Q5.

Q35: L 60-61: Given that the authors suggest that much earlier migrations shaped HBV diversity, it seems strange that they introduce the concept of dissemination of pathogens through globalisation and industrialisation which are processes that happened much later.

We have made the statement more precise. We intend to convey that modern globalization contributes significantly to the spread of diseases, and similarly, historical exchanges and connections between different geographic locations have also contributed to the dissemination of pathogens. Please see lines 61-62.

Q36: L 62 - 63: "Remarkably, HBV only infects humans and a few other primate species²⁶" Given that closely related viruses (that are also classified as HBV) are also infecting rodents, bats and other mammals, this statement is wrong.

Thank you very much for correcting my mistake, we have made the necessary changes in the text. Please see line 65.

Q37: L 64: Typo "... tightly linked to human..."

We apologize for the typo. We have made the correction. We have thoroughly reviewed the manuscript to ensure that there are no similar issues.

Q38: L 70: 10,000 isn't really a deep timescale.

We have made the modifications according to your suggestions. Please see lines 71-72.

Q39: L 72-75: Members of the WENBA lineage were already described (although not named as such) in <https://doi.org/10.1038/s41586-018-0097-z> and <https://doi.org/10.7554/eLife.36666.001>.

We have made the modifications according to your suggestions. Please see lines 74-76.

Q40: L 79: Typo "Only two genomes have been recovered."

We apologize for the typo. We have made the correction. We have thoroughly reviewed the manuscript to ensure that there are no similar issues.
Q41: L 96: Reference to ‘these positive samples’ where it is unclear which previously mentioned sample set is referred to.
We have made the modifications according to your suggestions. Please see lines 100-101.
Q42: L 105: The use of ‘impressive’ here is inappropriate.
We have made the modifications according to your suggestions.
Q43: L 106: The use of ‘successfully’ here is inappropriate.
We have made the modifications according to your suggestions.
Q44: L 109-111: This statement is out of place in the results and should be moved to the discussion.
We have made the modifications according to your suggestions.
Q45: L 115: The tree shows only 20 sequences, although the WENBA sequence may have been collapsed?
There are 25 sequences are included in phylogenetic analysis after we add 4 new sequences.
Q46: L 119: what does ‘more derived’ mean?
This word has been removed.
Q47: L 161: The reference to figure S4 seems inappropriate here.
We have made the correction.
Q49: L 186: This point is repeated from line 179. Also, I’m not an expert in interpreting ADMIXTURE results, but I don’t see a consistent trend in differences between the genetic profile for individuals with genotype B and D.
In the PCA (Principal Component Analysis) results, individuals that are physically closer to each other tend to also be genetically closer. We can observe that individuals with genotype B cluster together on the PCA, whereas individuals with genotype D are spaced far apart. In Admixture results, different colors represent different genetic components, and the proportion of each color indicates the proportion of these genetic components within an individual. If two individuals have similar genetic component proportions, it suggests that they are genetically close. Conversely, if there is a substantial difference in the genetic components and their proportions between two individuals, it indicates that they are genetically distant. From the Admixture results, we can see that individuals with genotype B possess three genetic components with roughly similar proportions. On the other hand, those carrying genotype D exhibit a greater variance in the number of genetic components they have, as well as in the proportions of these components. Therefore, we would say that the Admixture results are consistent with those of the PCA. We hope my response has answered your question.
Q50: L 190: ‘(HBV genome have been published in 2018)’
We have made the correction.
Q51: L 201: Reference 31 seems out of place here.
We have made the correction.
Q52: L233-238: It’s not clear what the ‘In contrast’ refers to. The discussion of genotype A origins does not tie in with any data that is presented and seems out of place.

In contrast’ refers to compared to the numerous HBV of genotype B we identified. Only one individual carrying genotype A was found in this study. we only identified one individual carrying genotype A. The subsequent discussion aims to illustrate that conclusions about the origins of genotype A based solely on modern genomic data may be somewhat one-sided, and the lack of ancient data prevents us from determining the geographical origin of genotype A with certainty.

Q53: L 234: Stray ‘S’ at the beginning of the line.

We apologize for our carelessness. We have made the correction.

Q54: L 246 - 254: This is largely a repeat from the results section.

We have removed this part.

Q55: L 256 and 258: The authors should be explicit with what they mean by ‘this region’.

We have changed “this region” into “Xinjaing”. Please see lines 301-303.

Q56: L 382: C->T changes occur at the 5’ end, not the 3’ end, G->A changes on the 3’ end.

We have made the correction.

Q57: L 399: The authors state that 14 out of 28 positive samples were radiocarbon dated, but the paper mentions 30 positive samples? Table S1 only gives radiocarbon dates for 13 samples. Why?

We have revised and clarified in the main text. Please see lines 102-105.

Q58: L 404-406: do the dates assigned from other individuals from the same site agree with the context that the non-radiocarbon dated samples were found in?

The background of individuals from the same sites is the same. And we state this fact in the main text.

Q59: L 408: The main text says that 21 sequences were included, not 20?

Please see Q45.

Q60: L 410: State which coverage depth was used for determining the 50% coverage.

We use the newly reconstructed ancient genomes that have over 50% genome coverage and a mean coverage greater than 5x.

Q61: L 413: What is A336? Give reference for that sample.

A336 is the old name of XHM16. We have made the correction.

Q62: L 420: Fig S4 doesn’t support the statement that is being made.

We have made the correction.

Q63: L 424: Fig S2 doesn’t support the statement that is being made.

We have made the correction.

Q64: L 444: What is the “1,240k panel”? Give reference. Archaeological background for Bayanbulag gives radiocarbon dates for two samples (AT6, AT8) that aren’t listed in the title of the section.

We have now provided the references to “1,240k panel”. The two samples, AT6 and AT8, were found in the same location within the burial site as AT7, and archaeologists believe that these individuals were buried at the same time. We screened all individuals from this site for HBV, but HBV reads were only detected in individuals AT7, AT19, and AT24. Although HBV was not detected in samples AT6 and AT8, their inclusion is significant for the chronological estimation of other samples. Therefore, we have listed this result in the background information section.

Reviewer #3 (Remarks to the Author):

The study describes sequencing data from 30 ancient HBV genomes dated between approximately 4,130 and 715 years ago from 13 sites in Eastern Eurasia. HBV is a globally distributed pathogen and the history of its infection in humans dates back 10,000 years. The results demonstrated that genotypes B, C and D may have originated in Eastern Asia. Furthermore, a high level of HBV diversity in a single location in Xinjiang, characterized by the presence of three different genotypes (A, B, D), underlines the importance of migrations and human interactions in the spread of HBV. Furthermore, the authors report the identification of a transition from non-recombinant subgenotypes (B1, B5) to recombinant subgenotypes (B2-B4). This suggests a change in epidemiological dynamics in Eastern Eurasia over time. The authors highlight that the study reveals the regional origins of the predominant genotypes and the changes in viral subgenotypes over the centuries.

We would like to extend my sincere gratitude for the valuable comments and suggestions you provided during the review of my manuscript. Your insights have been instrumental in enhancing not only the clarity and readability of the paper but also its scientific rigor and completeness. The thoroughness of your review has undoubtedly improved the quality of my work, and for this, we are truly appreciative.

Q1: In my opinion, the study is interesting and presents new ancient HBV sequencing data in samples from Eastern Eurasia. The methodology for obtaining old HBV data appears to be adequate. However, it is necessary to have criteria to validate the viral DNA data obtained from these very old samples. A major challenge for any study of very ancient DNA is the accumulation of post-mortem damage to the genome of interest. These damages include fragmentation, nucleotide deamination, and polymerase-blocking lesions, such as molecular cross-linking, resulting from enzymatic and chemical reactions. Predictably recurring forms of damage, such as the tendency for cytosine deamination to occur more frequently near the 3' and 5' ends of fragmented DNA molecules, also provide a means of addressing contamination issues and inferring the authenticity of a recovered sequence. through statistics. pattern analysis. Therefore, I suggest that the authors present more evidence of the authenticity of the analyzed DNA.

As you mentioned, the damage pattern at the ends of DNA fragments serves as the gold standard for verifying their ancient origin. Thus, based on the damage pattern of fragments from ancient humans in the sequencing library, we can determine that the library originates from ancient samples. This means that as long as we prevent modern HBV contamination, we can ensure that all HBV fragments are ancient. To avoid modern HBV contamination, we made sure that the experimental personnel were not infected with HBV, that the laboratory was clean, and that it was isolated from any labs handling modern HBV. Furthermore, during the high-throughput sequencing, none of the samples sequenced alongside ours were related to modern HBV. Finally, combining the observed damage pattern at the ends of the HBV fragments and their basal position in the phylogenetic analysis, we conclude that the ancient HBV sequences we recovered are authentic ancient sequences.

Q2: I am also concerned about the sequence dataset used (including old and new HBV DNA sequences) as well as the results of the phylogenetic and phylodynamic analyzes

presented. Methods that predict the temporal dynamics and phylogeography of recent virus emergence have been remarkably effective in reconstructing recent virus evolutionary histories. Extrapolating substitution rates to longer periods could provide the means to reconstruct much deeper evolutionary histories of viruses. However, a number of recent developments challenge the applicability of such methods to the widely accepted concepts of virus evolutionary timescales. Noteworthy, HBV is a partially double-stranded DNA virus with a circular genome of just 3,200 nucleotides. It employs error-prone reverse transcriptase (RT) in part of its replication process, making its evolutionary rate (substitution/site/year) closer to that of RNA viruses than DNA viruses. Furthermore, the four open reading frames (ORFs), which encode the surface (S), nucleocapsid (C), and X proteins and the polymerase (P), largely overlap, restricting viral evolution because synonymous mutations in one frame are often non-synonymous in another. Previous studies have already demonstrated a contrast between high replacement rates in the short term and apparently low replacement rates in the long term (rate mismatch). There is also a high diversity of intra-host HBV, creating even more difficulties in the analysis.

Your inquiry is immensely valuable to our research. In addressing the discrepancies in HBV evolutionary rates over short and long terms, we have utilized a substantial number of temporally informed samples for calibration. Among the 282 samples included in our phylogenetic analysis, 148 are ancient samples with established sample times, which are of significant importance for the calibration of the molecular clock. The inclusion of these ancient samples, coming from various time points and geographic locations, helps us rectify biases that may arise from uneven temporal and geographic sampling. In addition, we have rigorously tested our prior models and have employed the strictest model that is most suitable for our dataset. Collectively, these measures give us confidence in addressing some of the challenges associated with reconstructing the deep evolutionary history of HBV. We also look forward to incorporating more ancient samples from different times and locales in the future to further refine the deep evolutionary history of HBV.

Q3: In addition, the manuscript presents the classification of ancient HBV sequences using the current taxonomy of genotypes and subgenotypes. A previous scientific paper (cited by the authors) presented the occurrence of a West Eurasian Neolithic to Bronze Age (WENBA) HBV lineage. Why couldn't the same happen in Eastern Eurasia? In my opinion, all these questions should be answered and included in the manuscript.

Thank you for your question. There are two main reasons that might explain our current findings. First, the number of samples we have screened is not yet sufficiently large, and the temporal scale is not deep enough. The number of ancient samples spanning tens of thousands of years that we have examined is inadequate. Therefore, we have limited knowledge about the ancient HBV types on the Eastern Eurasian continent, as well as their distribution and infection patterns.

Second, unlike in Western Eurasia, where extensive population replacements and mixing events have occurred since 5000 years ago, the genetic composition of populations on the Eastern Eurasian continent has been relatively stable. This stability in human genetics could lead to a more consistent propagation and spread of HBV types across Eastern Eurasia without significant large-scale replacement events.

We recognize the need for a larger and more temporally extensive sample size to better understand the ancient HBV types and their dynamics. We also suggest that

studying the stability of genetic components in populations could provide valuable insights into the transmission patterns of HBV.

Finally, thank you again for your patience, and your suggestions and guidance for our work. These are all the changes we have made in response to your comments. We hope that we can meet the publication requirements of the journal, and we also hope that we can get your further suggestions and communication.

REVIEWER COMMENTS

Reviewer #1 (Remarks to the Author):

Thank you for making the suggested changes; the unrooted phylogenies are a useful addition.

Reviewer #2 (Remarks to the Author):

The authors present a revised version of the manuscript. It represents an improvement on the first version with regards to the analyses performed, but the language, reporting and presentation of data requires further revision.

Strong language/overinterpretation:

L 64: 'equivalent role' How do the authors know the role of ancient trade routes and human interactions is equivalent to industrialization and economic globalisation?

L 36: This point is made a number of times (e.g. L 256, L309), but presented in the abstract to a reader without background knowledge of the site, this is out of context. Furthermore, is a difference of two or three genotypes being present at a single site really that big, given how the individuals are dated? Have there been enough individuals screened and deemed positive from the same site in Western Eurasia to suggest that the presence of these three genotypes at a single site in Eastern Eurasia is surprising - I would be very surprised if this is the case? Regarding the discussion, I suggest that discussion of this point to be condensed in a single location, rather than mentioning it multiple parts.

L 254 - 257: I don't think the discovery of a WENBA sequence is surprising on the grounds that genotype G is absent in Eastern Eurasia. Genotype G could have evolved in Western Eurasia out of a diversity of WENBA sequences that includes sequences in Eastern Eurasia.

L 282 - L 286: This point has already been made by Mühlemann et al., as well as Kocher et al.

L 286 - 292: In general, the basal position of a sequence in a geographic region should not be interpreted as evidence of origin, since there may always be an older sequence recovered in the future. In this particular case, 98JJLM9 is reported to be a mixed infection, which may affect its position in the tree. If the authors want to speculate on the origin of genotype C based on this individual, the authors should present two (or multiple) consensus sequences corresponding to the two (or multiple) viruses putatively infecting the individual. Else I suggest removing this point.

L343 and following. It's not immediately clear to me whether 'no earlier than 1.8 kya' means that the recombination event has to have happened before 1.8 kya or after 1.8 kya. I also don't think that a claim can be made on the time interval when the recombination event happened since recombinant and non-recombinant lineages can co-circulate and there is vast undersampling of the ancient diversity.

L 369-372: The authors should refer to their own analyses.

L 349-360: This paragraph is a simplification. Why do the authors think that only substitutions in the S protein could affect infectivity? The virus is infinitely complex and mutations in multiple genes can affect infectivity or transmission.

Reporting / presentation of data

The way the data is presented and reported makes it difficult to follow what was done where, how and why. I mention some points in the sections below, but there may be more, so I suggest the authors carefully check the manuscript and SOM if there are any additional inconsistencies or improvements that can be made.

First, it is difficult to follow what data was included in which analyses and there are inconsistencies within the text:

Data S1-S5 is missing.

The first section in the results should state how many samples had how much coverage.

Reporting on genome coverage is discordant. Line 128 suggests consensus sequences were made at 5x coverage, line 113 reports 3x. Better be consistent.

L 461 suggests 21 sequences were included in an initial ML tree, but in the tree I count 26.

L 135: Fifteen genotype B samples. Then why does Table S2 contain 17 sequences?

The tree in figure S3b contains 17 sequences. $17 + 8$ mixed infections = 25, so here's another number of sequences that's different.

L 173: '... same set of sequences as for the ML tree' But there were two sets of sequences used to make the ML tree.

Second, there are discrepancies in the descriptions of what has been done between the Methods and the Results:

The Results suggest that an alignment with bwa against a HBV reference was done (L102), this is not described in the methods for the initial screening of the samples. When was the bwa alignment done?

L 100-102: This explanation of the capture process is out of place. At this point of reading the manuscript, the reader won't yet know that capture was performed.

L103-104: The way this sentence is worded implies that multiple samples were tested from each individual, but only one (petrous or tooth) yielded DNA. This should be reworded, to just state whether petrous bone or teeth were tested.

L114-116: This sentence doesn't make sense. What is meant by 'degraded'?

L119 -120: Say how these mixed infection samples were treated? Was there an attempt to tease apart the genomes from those two (or more) viruses? If not, how can you be sure that the consensus is correct? Give references to the previously employed methods that were used to identify the mixed infections.

Inconsistencies between explanations in Methods and Results (e.g. which trees were made - the Methods don't include the different ML trees that were made).

Methods on Temporal signal analysis and phylogenetic analysis need to state which ML tree was used as input for Tempest. The text should also include which model was used to make the final BEAST tree and how this was determined (path sampling etc).

Table S2: some of the identities that were calculated are very low (0.545). This looks wrong, since they would suggest that these sequences are highly divergent. Maybe the calculations were done including non-defined sites. This should be mentioned, or these sites ignored. Furthermore, L 142-148 states that there is 'high sequence identity' between some sequences. However, there is no discussion about whether this is relative to some other sequence, or what cut-offs were used.

L 176-177: 'additional phylogeny excluding individuals with mixed infections'. As far as I can tell, there's no dated phylogeny that includes all the mixed infections.

L 208-209: This analysis isn't described in the Methods, and no data are presented supporting it. I suggest the authors add this analysis.

The discussion is very disjointed with the same issue being discussed multiple times in different locations of the discussion. I suggest condensing those into a single location (e.g discussion of the WENBA sequence which is split into multiple locations (L 254-257/L327-328), discussion of the 'high diversity' in Xinjiang etc).

Third, the presentation of some of the results is unclear and doesn't allow interpretation:

Presentation of aDNA damage is insufficient, the individual samples in Figure S2A are not readable. The x-axis label is not readable. It would be better if this figure was a panel of figures for each individual, with the HBV and human damage patterns plotted as two curves. It could also include the number of HBV reads that are included for each individual.

The recombination figure is unreadable. Annotate with genotypes of the sequences and the recombination event that is shown. The colours on the grey background are difficult to read. RDP allows export of the inferred data for re-plotting to customise the figures and make them easier to read. Furthermore, the figure actually does not allow the reader to convince themselves that there is no recombination in the ancient genotype B sequences. Results should be shown for recombination analyses for these sequences specifically. The SimPlot analyses mentioned in the response should be shown in the manuscript. The discussion in the response regarding low coverage and its implication on the recombination analysis should be mentioned in the text.

Fig 3: The tree contains two different scales, the scale needs a label. The label for genotype J suggests that this is a clade full of genotype J, but the label on the collapsed clade suggests something else. The legend should indicate the abbreviations in the figure (e.g. ORU, CPZ etc). What do the numbers at the internal node represent? What are the blue bars?

L 504: Name the two sites with multiple genotypes on the map

L 510: How were the time period intervals chosen?

L 142-148: I cannot find matching support values in any of the ML trees. Please check and reference the actual figure that is being mentioned (since there are three ML trees in the

supplement).

Fourth, multiple reviewers have commented that the paper contains a lot of language that states that results were 'surprising' or 'notable'. While some of those instances have been removed, there are still remaining problems including L 33-34 (Appears that, Notably), L 39 (Unravels), L 88 (Plausible), L 88 (remarkably), L 134 (extremely long branch. There are other samples with similarly long branches), L 108 (Fortunately), L 245 (unprecedented), L 251 (remarkable), etc.

Reviewer comments that haven't been dealt with

L 94-95: This point has not been corrected.

L 75 ff: WENBA lineage: also found in earlier studies, just not named.

Reviewer 1 Q4: The wording in the abstract (L 36 of original abstract, L 36 in revised abstract) with regards to the shift in non-recombinant to recombinant genotypes hasn't been changed.

Reviewer 2 Q8: There is no description of how priors were chosen. This should be added to the methods.

Reviewer 2 Q14: I appreciate the additions that were made to the site descriptions, but the references to the previous studies are still part of the manuscript and do not support the point being made.

Minor comments

L 128: s/This/These

L 504: s/lineage/genotype

L 105: 'combining literature and...' this sounds strange. If literature was considered, this should be referenced, or at least reference the SOM.

L 148: This implies that the sequence was basal to all sequences. Be explicit that this is basal to genotype B.

Reviewer #3 (Remarks to the Author):

The study describes sequencing data of 30 ancient HBV genomes dating between approximately 4130 to 715 years ago sourced from 13 sites across Eastern Eurasia. HBV is a globally distributed pathogen and the history of HBV infection in humans predates 10,000 years. The results demonstrated that genotypes B, C, and D may have originated in Eastern Asia. Furthermore, a high level of HBV diversity at a single site in Xinjiang, characterized by the presence of three different genotypes (A, B, D), underscoring the significance of human migrations and interactions in the spread of HBV. Furthermore, the authors report the identification of a transition from non-recombinant subgenotypes (B1, B5) to recombinant subgenotypes (B2-B4). This suggests a shift in epidemiological dynamics within Eastern Eurasia over time. The authors highlight the study unravels the regional origins of prevalent genotypes and shifts in viral subgenotypes over centuries.

In accordance with my previous main questions about this manuscript, the authors have answered all of them: (1) the criteria for validating viral DNA data obtained from these very old samples; (2) the sequence dataset used (including old and new HBV DNA sequences) as well as the results of phylogenetic and phylodynamic analyses; (3) the classification of HBV sequences using the current taxonomy of genotypes and subgenotypes.

However, they did not modify the manuscript in accordance with some of these responses. I don't think it's mandatory. But I was particularly concerned because the manuscript presents analysis of ancient HBV genomes alongside recent ones. My additional suggestion would be to perform separate phylogenetic and phylodynamic analyses (mainly with temporal data) for each genotype to reinforce/refine previous findings (this could be highlighted in the Discussion, without presenting additional figures). I also consider it important to consider these issues in a paragraph on limitations and needs for new studies to better elucidate the phylogenetic classification and evolution of HBV in eastern Asia and other regions of the World.

Point-by-Point Response to Reviewer Comments for NCOMMS-23-44058A
“Origin and dispersal history of Hepatitis B virus in Eastern Eurasia”

We sincerely appreciate your meticulous reading of our manuscript and your professional and constructive suggestions. We are truly grateful for all your comments and suggestions, which have contributed to enhancing the quality of our manuscript. Following the recommendations of the three reviewers, we have made the necessary revisions to our manuscript. Our point-by-point responses to the reviewers' comments are highlighted in blue, and the changes made to the Main Text and Supplements are also appropriately marked. Additionally, for the convenience of the reviewers, we have provided a clean version of the supplementary folder.

Reviewer #2 (Remarks to the Author):

The authors present a revised version of the manuscript. It represents an improvement on the first version with regards to the analyses performed, but the language, reporting and presentation of data requires further revision.

Strong language/overinterpretation:

Q1: L 64: ‘equivalent role’ How do the authors know the role of ancient trade routes and human interactions is equivalent to industrialization and economic globalisation?
Considering the potential variations in transportation and other factors between ancient and modern times, we agree to remove this statement.

Q2: L 36: This point is made a number of times (e.g. L 256, L309), but presented in the abstract to a reader without background knowledge of the site, this is out of context. Furthermore, is a difference of two or three genotypes being present at a single site really that big, given how the individuals are dated? Have there been enough individuals screened and deemed positive from the same site in Western Eurasia to suggest that the presence of these three genotypes at a single site in Eastern Eurasia is surprising - I would be very surprised if this is the case? Regarding the discussion, I suggest that discussion of this point to be condensed in a single location, rather than mentioning it multiple parts.

In the discussion section, we have focused our discourse on the diversity of Eastern Eurasia. The sample size for Western Eurasia, nearing 150, is uniformly distributed across its various regions. Current data suggest the presence of only two HBV genotypes in Western Eurasia during the period 5000-3000 years ago, whereas Eastern Eurasia demonstrated a higher diversity with five distinct genotypes. Hence, we propose that the diversity of HBV was more pronounced in Eastern Eurasia during this specified timeline.

Q3: L 254 - 257: I don't think the discovery of a WENBA sequence is surprising on the grounds that genotype G is absent in Eastern Eurasia. Genotype G could have evolved in Western Eurasia out of a diversity of WENBA sequences that includes sequences in Eastern Eurasia.

We lend our support to the hypothesis suggesting the origin of genotype G in Western Eurasia, given that a majority, over 95%, of WENBA samples, originate from this area. Nonetheless, it's noteworthy that the contemporary prevalence of genotype G is scarce in Asia, particularly with no reported detections in China to date. This contrasts with the findings of WENBA in

ancient Eastern Eurasia, underscoring a notable discrepancy in HBV distribution across ancient and modern eras. In light of these observations, we have thoughtfully expanded upon this phenomenon in our analysis.

Q4: L 282 - L 286: This point has already been made by Mühlemann et al., as well as Kocher et al.

In the introduction, we have added a description regarding previously published research that has reported sequences belonging to the WENBA lineage.

Q5: L 286 - 292: In general, the basal position of a sequence in a geographic region should not be interpreted as evidence of origin, since there may always be an older sequence recovered in the future. In this particular case, 98JLM9 is reported to be a mixed infection, which may affect its position in the tree. If the authors want to speculate on the origin of genotype C based on this individual, the authors should present two (or multiple) consensus sequences corresponding to the two (or multiple) viruses putatively infecting the individual. Else I suggest removing this point.

Currently, it is not possible to reconstruct major and minor strains from data on mixed infections. Hence, we are unable to segregate the sequences of the individual 98JLM9. This may affect the topology of the tree. Considering that ancient HBV has only been discovered in Eastern Eurasia at present, we still believe that genotype C originated in Eastern Eurasia. However, science demands rigor, so we have removed the statements suggesting the origin of genotype C in Eastern Eurasia from this article. In the future, we hope to gather more evidence to explore the origins of genotype C.

Q6: L343 and following. It's not immediately clear to me whether 'no earlier than 1.8 kya' means that the recombination event has to have happened before 1.8 kya or after 1.8 kya. I also don't think that a claim can be made on the time interval when the recombination event happened since recombinant and non-recombinant lineages can co-circulate and there is vast undersampling of the ancient diversity.

In the article, we have revised the terminology to suggest that the recombination event may have occurred after the year 1800.

Q7: L 369-372: The authors should refer to their own analyses.

Thank you for pointing that out.

Q8: L 349-360: This paragraph is a simplification. Why do the authors think that only substitutions in the S protein could affect infectivity? The virus is infinitely complex and mutations in multiple genes can affect infectivity or transmission.

HBV is highly complex, and here we are merely pointing out one possibility. It is not to say that changes in the S protein will definitively affect infectivity. To make the expression more precise, we have already removed the descriptions related to the S protein.

Reporting / presentation of data

The way the data is presented and reported makes it difficult to follow what was done where, how and why. I mention some points in the sections below, but there may be more, so I suggest the authors carefully check the manuscript and SOM if there are any additional inconsistencies or improvements that can be made.

First, it is difficult to follow what data was included in which analyses and there are inconsistencies within the text:

Q9: Data S1-S5 is missing.

Now, we have placed data S1-S6 in the supplementary folder.

Q10: The first section in the results should state how many samples had how much coverage.

We have added a description of sample coverage in the first part of the results section.

Q11: Reporting on genome coverage is discordant. Line 128 suggests consensus were made at 5x coverage, line 113 reports 3x. Better be consistent.

Line 128 indicates that the samples used in the phylogenetic analysis must have a mean coverage greater than 5x, while line 113 refers to the percentage of sites with 3x coverage out of the total number of sites. Thus, these lines describe different aspects. To avoid reader confusion, we have revised line 113 to reflect the percentage of sites with 1x coverage out of the total number of sites.

Q12: L 461 suggests 21 sequences were included in an initial ML tree, but in the tree I count 26.

The number of sequences in line 461 is incorrect. Thank you for pointing this out. We have modified it in the main text. The number of samples added to the phylogenetic analysis is 25. We checked the tree and the fasta files used for tree construction, which all showed that the number of new ancient sequences added to the tree construction is 25.

Q13: L 135: Fifteen genotype B samples. Then why does Table S2 contain 17 sequences? The tree in figure S3b contains 17 sequences. $17 + 8$ mixed infections = 25, so here's another number of sequences that's different.

There are two sequences that have not been added to the phylogenetic analysis, but they have been analyzed for sequence identity. We have removed these two sequences. In addition, there is a mixed infection sample ZQM16 in Figure S3b, which has been removed and the maximum likelihood tree and MCC tree have been constructed again after removing the mixed infection samples. The number of mixed infection samples in the main text is also incorrect. The number of mixed infection samples should be nine. $16 + 9 = 25$, and we have modified it in the main text.

Q14: L 173: '... same set of sequences as for the ML tree' But there were two sets of sequences used to make the ML tree.

Two datasets were used for the construction of the MCC tree, the results included the mixed HBV infections displayed in Fig 3 and the MCC tree without mixed infections displayed in Fig S5a.

Second, there are discrepancies in the descriptions of what has been done between the Methods and the Results.

Q15: The Results suggest that an alignment with bwa against a HBV reference was done (L102), this is not described in the methods for the initial screening of the samples. When was the BWA alignment done?

We have already added this part in the method.

Q16: L 100-102: This explanation of the capture process is out of place. At this point of reading the manuscript, the reader won't yet know that capture was performed.

We have adjusted the order of descriptions in the results section.

Q17: L103-104: The way this sentence is worded implies that multiple samples were tested from each individual, but only one (petrous or tooth) yielded DNA. This should be reworded, to just state whether petrous bone or teeth were tested.

We do not analyze multiple samples from each individual. Our preferred target for analysis is teeth, and for samples without teeth, we use the temporal bone as the source for analysis. We have added a description of the source of the experimental samples in the results section of the main text.

Q18: L114-116: This sentence doesn't make sense. What is meant by 'degraded'?

To make the description more accurate we modified the sentence to “For the samples XBQM86 and XHM31, the capture experiment was unsuccessful, leading to a loss of DNA content post-capture compared to its pre-capture state.”

Q19: L119 -120: Say how these mixed infection samples were treated? Was there an attempt to tease apart the genomes from those two (or more) viruses? If not, how can you be sure that the consensus is correct? Give references to the previously employed methods that were used to identify the mixed infections.

Compared to normal individuals, those with mixed infections have a higher proportion of mixed sites. We assessed signals suggestive of heterozygosity throughout the genome and insertion events at the 5' end of the C gene³⁰. The frequencies of the major and minor mutations at each site are calculated and mixed sites are covered at least 10 times, with the major mutation frequency being less than 90%, and the minor mutation frequency greater than 10%. Mixed sites with a major mutation of G and a minor mutation of A, or a major mutation of C and a minor mutation of T, are excluded to ensure that the heterozygosity is not due to ancient DNA damage. Following these criteria, the number of mixed sites is counted, and the overall proportion of positions covered more than 10 times in the dataset that are detected as mixed is calculated. This value serves as the baseline for determining whether an infection is mixed. Previously, no studies had been conducted to separate the sequences of major and minor strain from mixed infection data simultaneously. Consistent with the methods used in previous ancient HBV studies, mixed sites are filtered during the construction of the consensus sequence, retaining only those sites with a frequency greater than 90%. This ensures that the consensus sequences we generate belong to the primary strain.

Q20: Inconsistencies between explanations in Methods and Results (e.g. which trees were made - the Methods don't include the different ML trees that were made). Methods on Temporal signal analysis and phylogenetic analysis need to state which ML tree was used as input for Tempest. The text should also include which model was used to make the final BEAST tree and how this was determined (path sampling etc).

We have already added this information in the method. We have conducted the path sampling analysis again. In the analysis, we used the parameters chosen by Arthur in his article published in Science, and based on the results of the reanalysis, we selected the best prior model and conducted the BEAST analysis anew.

Q21: Table S2: some of the identities that were calculated are very low (0.545). This looks wrong, since they would suggest that these sequences are highly divergent. Maybe the calculations were done including non-defined sites. This should be mentioned, or these sites ignored. Furthermore, L 142-148 states that there is 'high sequence identity' between some sequences. However, there is no discussion about whether this is relative to some other sequence, or what cut-offs were used.

Because positions without coverage were not excluded during the computation process, these uncovered positions were all considered as sites of diversity. We rerun this analysis with uncovered positions excluded. Please see Table S2. Also, we have added an analysis of sequence identity among ancient HBV genotype B, and this result can reflect the relationship between ancient HBV genotype B strains. Please see data S4.

Q22: L 176-177: ‘additional phylogeny excluding individuals with mixed infections’. As far as I can tell, there’s no dated phylogeny that includes all the mixed infections. **For constructing the MCC tree, two datasets were used. The results included the mixed HBV infections displayed in Fig 3 and the MCC tree without mixed infections displayed in Fig S5a.**

Q23: L 208-209: This analysis isn’t described in the Methods, and no data are presented supporting it. I suggest the authors add this analysis.

We have already added this information to the method.

Q24: The discussion is very disjointed with the same issue being discussed multiple times in different locations of the discussion. I suggest condensing those into a single location (e.g discussion of the WENBA sequence which is split into multiple locations (L 254-257/L327-328), discussion of the ‘high diversity’ in Xinjiang etc).

We have concentrated the discussion on the diversity of Eastern Eurasia in the discussion section.

Q25: Third, the presentation of some of the results is unclear and doesn’t allow interpretation: Presentation of aDNA damage is insufficient, the individual samples in Figure S2A are not readable. The x-axis label is not readable. It would be better if this figure was a panel of figures for each individual, with the HBV and human damage patterns plotted as two curves. It could also include the number of HBV reads that are included for each individual.

We have redrawn the figures, placing the damage patterns of HBV and humans from the same individual together. Additionally, the number of reads has been labeled in the graphs.

Q26: The recombination figure is unreadable. Annotate with genotypes of the sequences and the recombination event that is shown. The colours on the grey background are difficult to read. RDP allows export of the inferred data for re-plotting to customise the figures and make them easier to read. Furthermore, the figure actually does not allow the reader to convince themselves that there is no recombination in the ancient genotype B sequences. Results should be shown for recombination analyses for these sequences specifically. The SimPlot analyses mentioned in the response should be shown in the manuscript. The discussion in the response regarding low coverage and its implication on the recombination analysis should be mentioned in the text.

We have replotted the recombination figure and added the SimPlot results for the low-quality ancient genotype B samples to the supplement. Additionally, we have included a description of the sample quality for the recombination analysis in the main text.

Q27: Fig 3: The tree contains two different scales, the scale needs a label. The label for genotype J suggests that this is a clade full of genotype J, but the label on the collapsed clade suggests something else. The legend should indicate the abbreviations in the figure (e.g. ORU, CPZ etc). What do the numbers at the internal node represent? What are the blue bars?

We have added labels in Fig 3 and revised the names of the labels for the collapsed branches. Additionally, the figure caption now explains that the numbers on internal nodes represent the priors of the branches, and the blue bars indicate the 95% confidence interval for the most recent common ancestor.

Q28: L 504: Name the two sites with multiple genotypes on the map

We have added the names of multiple sites on the map.

Q29: L 510: How were the time period intervals chosen?

To compare the diversity of ancient HBV across Eastern and Western Eurasia, time intervals were established following the guidelines outlined in Kocher et al. (2021).

Q30: L 142-148: I cannot find matching support values in any of the ML trees. Please check and reference the actual figure that is being mentioned (since there are three ML trees in the supplement).

We have verified the values of branch credibility and made corresponding adjustments according to Fig S3a.

Q31: Fourth, multiple reviewers have commented that the paper contains a lot of language that states that results were ‘surprising’ or ‘notable’. While some of those instances have been removed, there are still remaining problems including L 33-34 (Appears that, Notably), L 39 (Unravels), L 88 (Plausible), L 88 (remarkably), L 134 (extremely long branch. There are other samples with similarly long branches), L 108 (Fortunately), L 245 (unprecedented), L 251 (remarkable), etc.

We checked the main text and removed similar vocabulary.

Reviewer comments that haven’t been dealt with

Q32: L 94-95: This point has not been corrected.

We have replaced the citation in this sentence with references to literature involving samples mentioned in this paper.

Q33: L 75 ff: WENBA lineage: also found in earlier studies, just not named.

For constructing the MCC tree, two datasets were used. The results for MCC tree including the mixed HBV infections IS shown in Fig 3 and the MCC tree without mixed HBV infections is shown in Fig S5a.

Q34: Reviewer 1 Q4: The wording in the abstract (L 36 of original abstract, L 36 in revised abstract) with regards to the shift in non-recombinant to recombinant genotypes hasn’t been changed.

In the previous version, we replaced “identity” with “suggest”. In the current version, we modify the sentence to “Our results suggest the possibility of a transition from non-recombinant subgenotypes (B1, B5) to recombinant subgenotypes (B2-B4).” We suggest the possibility of recombination events occurring, allowing for the existence of other possibilities.

Q35: Reviewer 2 Q8: There is no description of how priors were chosen. This should be added to the methods.

In the methods section, we have described the parameters chosen for the priors.

Q36: Reviewer 2 Q14: I appreciate the additions that were made to the site descriptions, but the references to the previous studies are still part of the manuscript and do not support the point being made.

Many samples are unpublished, hence we referenced background information from sites with literature citations. Minor comments

Q37: L 128: s/This/These

We have made modifications.

Q38: L 504: s/lineage/genotype

We have made modifications.

Q39: L 105: ‘combining literature and...’ this sounds strange. If literature was considered, this should be referenced, or at least reference the SOM.

The chronological information for some samples was inferred based on references, hence we have also cited the corresponding literature following this statement.

Q40: L 148: This implies that the sequence was basal to all sequences. Be explicit that this is basal to genotype B.

Based on current results, this sample appears to be the basal lineage of HBV genotype B.

Reviewer #3 (Remarks to the Author):

The study describes sequencing data of 30 ancient HBV genomes dating between approximately 4130 to 715 years ago sourced from 13 sites across Eastern Eurasia. HBV is a globally distributed pathogen and the history of HBV infection in humans predates 10,000 years. The results demonstrated that genotypes B, C, and D may have originated in Eastern Asia. Furthermore, a high level of HBV diversity at a single site in Xinjiang, characterized by the presence of three different genotypes (A, B, D), underscoring the significance of human migrations and interactions in the spread of HBV. Furthermore, the authors report the identification of a transition from non-recombinant subgenotypes (B1, B5) to recombinant subgenotypes (B2-B4). This suggests a shift in epidemiological dynamics within Eastern Eurasia over time. The authors highlight the study unravels the regional origins of prevalent genotypes and shifts in viral subgenotypes over centuries.

In accordance with my previous main questions about this manuscript, the authors have answered all of them: (1) the criteria for validating viral DNA data obtained from these very old samples; (2) the sequence dataset used (including old and new HBV DNA sequences) as well as the results of phylogenetic and phylodynamic analyses; (3) the classification of HBV sequences using the current taxonomy of genotypes and subgenotypes.

Q1: However, they did not modify the manuscript in accordance with some of these responses. I don't think it's mandatory. But I was particularly concerned because the manuscript presents analysis of ancient HBV genomes alongside recent ones. My additional suggestion would be to perform separate phylogenetic and phylodynamic analyzes (mainly with temporal data) for each genotype to reinforce/refine previous findings (this could be highlighted in the Discussion, without presenting additional figures). I also consider it important to consider these issues in a paragraph on limitations and needs for new studies to better elucidate the phylogenetic classification and evolution of HBV in eastern Asia and other regions of the World.

Thank you for your suggestion. We conducted phylogenetic analysis for each genotype, and the results are largely consistent with the topology of the previously constructed evolutionary tree that includes all genotypes. We have placed the result figure in the supplement.

REVIEWERS' COMMENTS

Reviewer #2 (Remarks to the Author):

This paper describes a total of 34 HBV ancient HBV sequences from Eastern Eurasia and provide a valuable addition to already available ancient HBV sequences mainly from Western Eurasia. Most of my previous suggestions have been addressed, thank you. There are some additional points that should be corrected:

Q1/L62: The original wording of this section was: 'Similarly, ancient trade routes and human interactions played an equivalent role in the spread of diseases' The wording in the new version is 'Ancient trade routes and human interactions played an equivalent role in the spread of diseases' I consider these to be equivalent in meaning. Therefore, this comment has not been addressed.

The legend of figure S2 should be adapted to reflect the new figure. The results text refers to figures S2a and S2b, even though there's now only one figure. There are some panels in figure S2 where thousands of reads mapped to HBV, yet there doesn't appear to be a clear damage pattern (XN12, AT19, FLTM48, AT24, FLTM101, XBQM46, TJZM25-2). Are the damage patterns from UDG treated libraries? If so, the damage patterns should be shown only for the non UDG treated libraries, if possible. The figure should include which samples were sequenced with UDG treatment.

Figure S7 lacks a legend explaining the two panels. I also suggest removing the file path from the title of fig. S7B and adjusting the y axis so the data are shown in full. Also, it appears that this figure shows the SimPlot analysis for only one sequence, even though there should be more. Those should be added. Figure S7a needs axis labels. Furthermore, the top right in S7a doesn't appear to show a recombination event, as there is no cross-over between the different sequences. Similarly, the signal in the bottom right is low.

L108-L111: Since analysis with bwa and MALT isn't described in the results, maybe this text would fit better in the Methods.

L112: 'an HBV reference sequence': Say which one, or specify where this can be looked up.

L113: Suggests 7 sequences had 100% coverage, while the abstract suggests 10 sequences were complete. Which one is correct?

L151: 'The ancient sequences XHM18 has the same sequence identity with modern subgenotype B1 and B5 (Table. S2).' Table S2 doesn't seem to contain a B5 sequence.

L204-217: Figure S7a does not include plots that show the recombination analysis for the ancient genotype B and C sequences. The recombination event for genotype I should be included in the figure. The authors should provide citations to the previous research that has found similar results. The supplementary dataset containing the recombination analysis results ends in *.unk and it's not clear how it can be opened. I suggest that the data that is in it is included as figures in the supplementary materials. Data from data_S6 could also be shown as a figure in the supplement.

L212: 'Samples predating 1800, as well as...' Specify what '1800' refers to (years into the past? 1800 CE?).

L151-152: In fig. S5a, XHM18 and XBQM47 appear to be identical. How come the sequence identity is quite different? Were positions masked? And if so, why was a different alignment used for the nucleotide identity calculation?

Reviewer #3 (Remarks to the Author):

The authors revised and improved the manuscript, addressing my main concerns. I have no more questions and requests for improvements to the article.

Point-by-Point Response to Reviewer Comments for NCOMMS-23-44058B
“Origin and dispersal history of Hepatitis B virus in Eastern Eurasia”

We sincerely appreciate your meticulous reading of our manuscript and your professional and constructive suggestions. We are truly grateful for all your comments and suggestions, which have contributed to enhancing the quality of our manuscript. Following the recommendations of the three reviewers, we have made the necessary revisions to our manuscript. Our point-by-point responses to the reviewers' comments are highlighted in blue, and the changes made to the Main Text and Supplements are also appropriately marked. Additionally, for the convenience of the reviewers, we have provided a clean version of the supplementary folder.

Reviewer #2 (Remarks to the Author):

This paper describes a total of 34 HBV ancient HBV sequences from Eastern Eurasia and provide a valuable addition to already available ancient HBV sequences mainly from Western Eurasia. Most of my previous suggestions have been addressed, thank you. There are som additional points that should be corrected:

Q1: L62: The original wording of this section was: ‘Similarly, ancient trade routes and human interactions played an equivalent role in the spread of diseases‘ The wording in the new version is ‘Ancient trade routes and human interactions played an equivalent role in the spread of diseases‘ I consider these to be equivalent in meaning. Therefore, this comment has not been addressed.

Thank you very much for your comments, we have already deleted this sentence in the main text.

Q2: The legend of figure S2 should be adapted to reflect the new figure. The results text refers to figures S2a and S2b, even though there’s now only one figure. There are some panels in figure S2 where thousands of reads mapped to HBV, yet there doesn’t appear to be a clear damage pattern (XN12, AT19, FLTM48, AT24, FLTM101, XBQM46, TJZM25-2). Are the damage patterns from UDG treated libraries? If so, the damage patterns should be shown only for the non UDG treated libraries, if possible. The figure should include which samples were sequenced with UDG treatment.

Thank you very much for pointing it out. We have now marked the information regarding whether the samples were UDG-treated in the figure.

Q3: Figure S7 lacks a legend explaining the two panels. I also suggest removing the file path from the title of fig. S7B and adjusting the y axis so the data are shown in full. Also, it appears that this figure shows the SimPlot analysis for only one sequence, even though there should be more. Those should be added. Figure S7a needs axis labels. Furthermore, the top right in S7a doesn't appear to show a recombination event, as there is no cross-over between the different sequences. Similarly, the signal in the bottom right is low.

We have added a legend for Fig S7 and have swapped the order of Figures S7a and S7b. The paths previously appearing in the figure have been removed, the font size of the Y-axis has been adjusted, and simplot results for additional HBV samples have been included. Regarding the RDP result charts, we originally presented outcomes from two different methods, which might have led readers to

mistakenly infer the absence of recombination signals. To minimize misunderstanding, we have replaced the RDP charts with results obtained using the same method and added labels at the axis positions.

Q4: L108-L111: Since analysis with bwa and MALT isn't described in the results, maybe this text would fit better in the Methods.

Thank you for your comments; we have now included this sentence in the methods section.

Q5: L112: 'an HBV reference sequence': Say which one, or specify where this can be looked up.

We have now indicated the position of the reference sequence in the supplementary file at the end of the sentence.

Q6: L113: Suggests 7 sequences had 100% coverage, while the abstract suggests 10 sequences were complete. Which one is correct?

The description in the abstract is correct; we have amended the incorrect numbers.

Q7: L151: 'The ancient sequences XHM18 has the same sequence identity with modern subgenotype B1 and B5 (Table. S2). ' Table S2 doesn't seem to contain a B5 sequence.

B6 and the B5 refer to the same sequence, but there was a naming error in the documentation. We have corrected this mistake in Table S2.

Q8: L204-217: Figure S7a does not include plots that show the recombination analysis for the ancient genotype B and C sequences. The recombination event for genotype I should be included in the figure. The authors should provide citations to the previous research that has found similar results. The supplementary dataset containing the recombination analysis results ends in *.unk and it's not clear how it can be opened. I suggest that the data that is in it is included as figures in the supplementary materials. Data from data_S6 could also be shown as a figure in the supplement.

We have revised the cited figure of recombination event for genotype B and C. We have included the recombination results of genotype I. And we have cited previously published articles that has found similar results. Data S6 contains the results of the RDP analysis and requires the RDP software to open. Due to the large number of sequences involved, it is not feasible to display them as images. Therefore, we have chosen to add these results in the supplementary materials.

Q9: L212: 'Samples predating 1800, as well as...' Specify what '1800' refers to (years into the past? 1800 CE?).

Yes, the "1800" here refers to 1800 years ago, and we have modified it in the main text.

Q10: L151-152: In fig. S5a, XHM18 and XBQM47 appear to be identical. How come the sequence identity is quite different? Were positions masked? And if so, why was a different alignment used for the nucleotide identity calculation?

We used the same sequence for sequence identity and phylogenetic tree construction. During the sequence identity analysis, we did not include sites with no coverage. The XBQM47 sample has relatively lower coverage, which may result in the absence of some specific sites, leading to a significant difference in sequence identity results compared to XHM18.